**Data Availability Statement:** All relevant data are within the manuscript and in S5 and S6 Tables.

**Funding:** This research was supported by the Energy Biosciences Institute, USDA HATCH project

# Managing flowering time in *Miscanthus* and sugarcane to facilitate intra- and intergeneric crosses

Hongxu Dong[1¤], Lindsay V. Clark[1], Xiaoli Jin[2], Kossonou Anzoua[3], Larisa Bagmet[4], Pavel Chebukin[4], Elena Dzyubenko[4], Nicolay Dzyubenko[4], Bimal Kumar Ghimire[5], Kweon Heo[6], Douglas A. Johnson[7], Hironori Nagano[3], Andrey Sabitov[4], Junhua Peng[8], Toshihiko Yamada[3], Ji Hye Yoo[6], Chang Yeon Yu[6], Hua Zhao[9], Stephen P. Long[1], Erik J. Sacks[1] *

1 Department of Crop Sciences, University of Illinois at Urbana-Champaign, Urbana, Illinois, United States of America, 2 Department of Agronomy, Zhejiang University, Hangzhou, Zhejiang, China, 3 Field Science Center for Northern Biosphere, Hokkaido University, Sapporo, Hokkaido, Japan, 4 Vavilov All-Russian Institute of Plant Genetic Resources, St. Petersburg, Russian Federation, 5 Konkuk University, Neungdong-ro, Gwangjin-gu, South Korea, 6 Department of Applied Plant Sciences, Kangwon National University, Chuncheon, South Korea, 7 USDA-ARS Forage and Range Research Lab, Utah State University, Logan, Utah, United States of America, 8 HuaZhi Biotechnology Institute, Changsha Hunan, China, 9 College of Plant Science and Technology, Huazhong Agricultural University, Wuhan, China

¤ Current address: Department of Plant and Soil Sciences, Mississippi State University, Starkville, Mississippi, United States of America
* esacks@illinois.edu

## Abstract

*Miscanthus* is a close relative of *Saccharum* and a potentially valuable genetic resource for improving sugarcane. Differences in flowering time within and between *Miscanthus* and *Saccharum* hinders intra- and interspecific hybridizations. A series of greenhouse experiments were conducted over three years to determine how to synchronize flowering time of *Saccharum* and *Miscanthus* genotypes. We found that day length was an important factor influencing when *Miscanthus* and *Saccharum* flowered. Sugarcane could be induced to flower in a central Illinois greenhouse using supplemental lighting to reduce the rate at which days shortened during the autumn and winter to 1 min d$^{-1}$, which allowed us to synchronize the flowering of some sugarcane genotypes with *Miscanthus* genotypes primarily from low latitudes. In a complementary growth chamber experiment, we evaluated 33 *Miscanthus* genotypes, including 28 *M. sinensis*, 2 *M. floridulus*, and 3 *M. ×giganteus* collected from 20.9˚ S to 44.9˚ N for response to three day lengths (10 h, 12.5 h, and 15 h). High latitude-adapted *M. sinensis* flowered mainly under 15 h days, but unexpectedly, short days resulted in short, stocky plants that did not flower; in some cases, flag leaves developed under short days but heading did not occur. In contrast, for *M. sinensis* and *M. floridulus* from low latitudes, shorter day lengths typically resulted in earlier flowering, and for some low latitude genotypes, 15 h days resulted in no flowering. However, the highest ratio of reproductive shoots to total number of culms was typically observed for 12.5 h or 15 h days. Latitude of origin was significantly associated with culm length, and the shorter the days, the stronger the relationship. Nearly all entries achieved maximal culm length under the 15 h treatment, but the nearer to the equator an accession originated, the less of a difference in culm length

ILLU-802-311, and the DOE Office of Science, Office of Biological and Environmental Research (BER), grant nos. DE-SC0016264, and DE-SC0018420 (Center for Advanced Bioenergy and Bioproducts Innovation) awarded to EJS. The funders had no role in study design, data collection and analysis, decision to publish, or preparation of the manuscript.

**Competing interests:** The authors have declared that no competing interests exist.

between the short-day treatments and the 15 h day treatment. Under short days, short culms for high-latitude accessions was achieved by different physiological mechanisms for *M. sinensis* genetic groups from the mainland in comparison to those from Japan; for mainland accessions, the mechanism was reduced internode length, whereas for Japanese accessions the phyllochron under short days was greater than under long days. Thus, for *M. sinensis*, short days typically hastened floral induction, consistent with the expectations for a facultative short-day plant. However, for high latitude accessions of *M. sinensis*, days less than 12.5 h also signaled that plants should prepare for winter by producing many short culms with limited elongation and development; moreover, this response was also epistatic to flowering. Thus, to flower *M. sinensis* that originates from high latitudes synchronously with sugarcane, the former needs day lengths >12.5 h (perhaps as high as 15 h), whereas that the latter needs day lengths <12.5 h.

## Introduction

*Miscanthus* is an emerging bioenergy biomass crop in North America and Europe [1, 2]. As a $C_4$ perennial grass, *Miscanthus* is native to eastern Asia and Oceania from tropical to cold-temperate environments [3]. However, currently only one single triploid clone of *M. ×giganteus*, which is an interspecific hybrid between *M. sinensis* and *M. sacchariflorus*, is widely available for commercial production and new hybrids are needed. Additionally, *Miscanthus* is a close relative to *Saccharum* and is potentially a valuable genetic resource for improving sugarcane [4–8].

Control of flowering time is important to plant breeders because it allows them to make crosses of their choosing. Constraints on which genotypes can be used as parents in crosses would be severe impediments to plant improvement. Synchronization of flowering time between sugarcane and *Miscanthus* is necessary for making intergeneric crosses between these two species, because, like most warm-season grasses, pollen of these two genera quickly loses viability within the first ~2 h of dehiscence under typical growing conditions [9–12]. Moreover, because *Saccharum* and *Miscanthus* pollen is typically intolerant of desiccation, it is not readily stored frozen; thus, consistently effective and long-term pollen-storage methods have not yet been developed for these genera.

*M. sacchariflorus* has been considered a quantitative short-day plant [13], similar to sorghum and sugarcane. *M. sinensis* was described as day neutral by Deuter [14], whereas Jensen *et al.* [15] reported that flowering time in *M. sinensis* was more complicated, depending on multiple factors, including thermal time, temperature, photoperiod, and precipitation. In the field at Urbana, *M. sacchariflorus* flowers as early as July and as late as early November, whereas *M. sinensis* flowers from late July to mid-October [16, 17]. In sugarcane, floral initiation is induced by a small decrease (30–60 sec per day) in day length from about 12.5 h [18, 19]. Most sugarcane varieties need between 12 and 12.5 h of photoperiod to induce flowering [20–22]. In our greenhouses at Urbana, Illinois, flowering of diverse *Miscanthus* accessions typically is greatest from August through October and again from March through June. For most sugarcane breeding programs in the U.S., peak flowering is in November and December. In central Illinois, the rapid decrease in day length during the autumn is not conducive to flowering sugarcane plants in the greenhouse. Thus, it would be desirable to develop methods to synchronize the flowering time of *Miscanthus* and *Saccharum*, thereby facilitating the

introgression of desirable genes for improving sugarcane. Additionally, it would be advantageous to be able to better predict and control flowering time in *Miscanthus* so that we can more readily make crosses between different *Miscanthus* genotypes.

In this study, we conducted one set of experiments to explore the feasibility of synchronizing flowering time of *Saccharum* and *Miscanthus* in a central Illinois (~40˚ N) greenhouse, and a complementary experiment in growth chambers to understanding how day length impacts flowering time and plant growth of *M. sinensis*. The objectives were: 1) to assess the diversity of flowering time for *Miscanthus* and *Saccharum*, 2) to determine the effects of cultural treatments that we hypothesized could delay flowering time in *Miscanthus*, 3) to determine how day length in controlled environment chambers affects flowering time of *M. sinensis* accessions that originate from different latitudes.

## Materials and methods

### Experiment 1: Flowering time management of *Miscanthus* and sugarcane in a greenhouse

To determine how to synchronize the flowering of *Miscanthus* and *Saccharum*, a series of greenhouse experiments were conducted over three years (2014–2017; Expts. 1a-c). A key component of the study was to assess the diversity of flowering times within each genus, when plants were grown in a greenhouse at Urbana, IL under a photoperiod treatment that was expected to be conducive to flowering of sugarcane. We also evaluated cultural treatments that we hypothesized had the potential to delay flowering of *Miscanthus*, such as 4 ˚C cold storage to delay the start of growth, cutting plants to 15 cm above the soil surface, and the combination of cutting followed by one month of 4 ˚C cold storage.

A panel of 23 *Miscanthus* (Table 1) and 31 *Saccharum* accessions (Table 2) were studied. All plants were grown in a tall (6.1 m eave height), controlled-environment greenhouse at the University of Illinois Energy Farm at Urbana, IL (40.1˚ N, 88.2˚ W), located where there was no light pollution (e.g. from street lamps or buildings) that could interfere with the short-day treatment required to flower sugarcane. When natural day length reached 12.5 h in Urbana (14 September in 2014, 2015, 2016), supplemental light (MH 1000W/U/BT37 metal halide bulbs, Venture Lighting, Twinsburg, OH, US) was provided to decrease the day length by 1 min d$^{-1}$ until a photoperiod of 11 h d$^{-1}$ was reached (13 December in 2014, 2015, 2016), at which point the day length was held constant until exceeded by the natural day length (22 February in 2014, 2015, 2016). Additionally, in the third year experiment (2016–2017), we grew a second set of the *Miscanthus* genotypes in a greenhouse on the University of Illinois main campus (<5 km from the Energy Farm greenhouse), in which the plants were given constant 13 h d$^{-1}$ photoperiod, starting on 2 September until natural day length exceeded this value on 9 April. In the greenhouses, temperature during the day was maintained between 27–31 ˚C and at night temperature was between 22–26 ˚C. *Miscanthus* plants were grown in 7 L pots (T.O. Plastics, Clearwater, MN, USA) containing peat-based potting mix (Metro-Mix 900, Sun Gro Horticulture, Agawam, MA, USA), whereas the larger-growing *Saccharum* plants were grown in 17 L pots. Slow release fertilizer was applied to each pot (Osmocote Pro 17-5-11, 6 months; 35 g per 7 L pot and 140 g per 17 L pot; ICL Specialty Fertilizers, Dublin, OH, USA). Drip irrigation was supplied to each pot automatically twice per day. For each pot of *Miscanthus* and *Saccharum* studied, data was recorded weekly when a plant was actively flowering (newly opened florets dehiscing pollen).

The 2014–2015 greenhouse experiment (Expt. 1a) was initiated from 25 March to 21 April 2014. For each of 23 *Saccharum* genotypes, 1–8 pots were established from stem cuttings (Table 2). For each of 10 *Miscanthus* genotypes, 36 pots were established from divisions of

**Table 1. *Miscanthus* genotypes included in a study, conducted in Urbana, IL over three years, on how cultural management treatments of greenhouse-grown plants affects flowering time.**

| Entry | Ploidy | Lat. | Long. | Number of pots | | | | | | | | |
| | | | | 2014–2015† | | | 2015–2016‡ | | | 2016–2017§ | | |
| | | | | Control | Cut only | Cut plus 4°C cold | Control | Single rhizome planting | Cold storage pot division | Control | 1 min $d^{-1}$ decreasing day length | Constant 13 h $d^{-1}$ day length |
|---|---|---|---|---|---|---|---|---|---|---|---|---|
| *M. ×giganteus* 'Illinois-6x.06 (M×g2x-6)' | 6x | | | 6 | 15 | 15 | 3 | | | 3 | | |
| *M. floridulus* 'US56-0022-03' | 2x | -20.9 | 165.3 | 6 | 15 | 15 | 3 | | | 3 | | |
| *M. sacchariflorus* 'PMS-075' | 2x | 40.1 | 116.2 | 6 | 15 | 15 | 3 | | | 3 | | |
| *M. sacchariflorus* ssp. *lutarioriparius* 'PF30022' | 2x | | | 6 | 15 | 15 | 3 | | | 3 | | |
| *M. sacchariflorus* 4x 'Gifu-2010-008' | 4x | 35.4 | 136.8 | 6 | 15 | 15 | 3 | 12 | 8 | 3 | 9 | 9 |
| *M. sacchariflorus* 4x 'PF30153' | 4x | | | 6 | 15 | 15 | 3 | 12 | 8 | 3 | 9 | 9 |
| *M. sacchariflorus* 4x 'Tōhoku-2010-034' | 4x | 38.7 | 139.9 | 6 | 15 | 15 | 3 | | | 3 | | |
| *M. sinensis* 'PMS-204' | 2x | 31.7 | 114.9 | 6 | 15 | 15 | 3 | | | 3 | | |
| *M. sinensis* 'PMS-375' | 2x | 19.6 | 110.3 | 6 | 15 | 15 | 3 | | | 3 | | |
| *M. sinensis* 'PMS-436' | 2x | 41.3 | 123.7 | 6 | 15 | 15 | 3 | | | 3 | | |
| *M. sacchariflorus* 'RU2012-016' | 2x | 47.2 | 134.4 | | | | 3 | 12 | 8 | 3 | 9 | 9 |
| *M. sacchariflorus* 'RU2012-037' | 2x | 49.1 | 136.5 | | | | 3 | 12 | 8 | 3 | 9 | 9 |
| *M. sacchariflorus* 'RU2012-050' | 2x | 48.9 | 136.2 | | | | 3 | 12 | 8 | 3 | 9 | 9 |
| *M. sacchariflorus* 'RU2012-078' | 2x | 48.7 | 133.0 | | | | 3 | 12 | 8 | 3 | 9 | 9 |
| *M. sacchariflorus* 'RU2012-112' | 2x | 48.6 | 133.9 | | | | 3 | 12 | 8 | 3 | 9 | 9 |
| *M. sacchariflorus* 'RU2012-120' | 2x | 48.6 | 134.4 | | | | 3 | 12 | 8 | 3 | 9 | 9 |
| *M. sacchariflorus* 4x 'Gifu-2010-024' | 4x | 35.6 | 137.0 | | | | 3 | 12 | 8 | 3 | 9 | 9 |
| *M. sacchariflorus* 4x 'JM11-019' | 4x | 35.1 | 132.3 | | | | 3 | 12 | 8 | 3 | 9 | 9 |
| *M. sacchariflorus* 4x 'JM11-040' | 4x | 34.8 | 132.9 | | | | 3 | 12 | 8 | 3 | 9 | 9 |
| *M. sacchariflorus* 4x 'PF30157' | 4x | | | | | | 3 | 12 | 8 | 3 | 9 | 9 |
| *M. sacchariflorus* 4x 'Tōhoku-2010-025' | 4x | 39.7 | 140.2 | | | | 3 | 12 | 8 | 3 | 9 | 9 |
| *M. sacchariflorus* 4x 'Tōhoku-2010-036' | 4x | 38.4 | 140.3 | | | | 3 | 12 | 8 | 3 | 9 | 9 |
| *M. sacchariflorus* 4x 'Tōhoku-2010-037' | 4x | 38.4 | 140.3 | | | | 3 | 12 | 8 | 3 | 9 | 9 |

In each year (2014–2016), plants were grown in a greenhouse that provided decreasing day length of 1 min per day via supplemental light from high intensity discharge (HID) lamps starting when natural day length reached 12.5 h in Urbana, IL (14 September) until day length reached 11 h (13 December), then held constant until natural day length exceeded this value on 22 February. In 2016, an additional set of *Miscanthus* plants were also grown in a second greenhouse at Urbana, IL, in which day length was held at a constant 13 h via supplemental HID lamps, starting on 2 September until natural day length exceeded this value on 9 April.

† Pots of *Miscanthus* were established on 21 April 2014. Each of the 10 genotypes had six control pots that were grown in the greenhouse without any further treatments. *Miscanthus* treatments included 1) cutting plants ~15 cm above the soil and allowing them to immediately regrow, 2) cutting the plants and storing them for 1 month at 4 °C before returning them to the greenhouse to regrow, and 3) uncut controls. For the 10 *Miscanthus* genotypes, each of the cut and cut plus cold treatments was applied to 3 pots monthly from September to January. Empty cells indicate genotypes that were not included for specific year's experiment.

‡ Treatments were 1) stored rhizomes (planted every 4 weeks starting on 3 June 2015), 2) divisions of stored pots (planted every 4 weeks starting on 3 June 2015), and 3) controls (actively growing plants cut ~15 cm above the soil surface on 3 June). Each of the 23 genotypes had three control pots. For 15 selected *M. sacchariflorus* genotypes, three pots of single rhizome planting and two pots of cold storage pot division were made for each genotype monthly from June to September.

§ Pots of *Miscanthus* were established on 29 July 2016. Each of the 23 genotypes had three control pots. Control pots cut ~15 cm above the soil surface were compared with a set of pots stored at 4 °C and returned to the greenhouse at 4-week intervals from September to November. Each of the 23 genotypes had three control pots. For 15 selected *M. sacchariflorus* genotypes, six stored divisions per *Miscanthus* genotype were removed from cold storage monthly and three of these were planted in a greenhouse with 1 min $d^{-1}$ decreasing photoperiod protocol and another three divisions were planted in another greenhouse with a constant 13 h $d^{-1}$ day length.

**Table 2. Sugarcane and intergeneric hybrid genotypes included in a study of flowering time management in a greenhouse at Urbana, IL over three years.**

| Entry | Accession | Number of pots | | |
|---|---|---|---|---|
| | | 2014–2015 | 2015–2016 | 2016–2017 |
| *Miscanthus* × *Saccharum officinarum* 'Fiji 17' | PI212268 | | | 2 |
| *Miscanthus* × *Saccharum officinarum* 'Fiji 53' | PI271853 | | | 2 |
| *Miscanthus* × *Saccharum officinarum* 'Fiji 54' | PI268060 | | | 2 |
| *Miscanthus* × *Saccharum officinarum* 'Fiji 55' | PI271854 | | | 2 |
| *Miscanthus* × *Saccharum officinarum* 'Fiji 57' | PI276960 | | | 2 |
| *Miscanthus* × *Saccharum officinarum* 'Fiji 59' | PI268061 | | | 2 |
| *Miscanthus* × *Saccharum officinarum* 'Raiatea' | Q37075 | 8 | | |
| *Saccharum* hybr. 'CP14-1613' | CP14-1613 | | | 2 |
| *Saccharum* hybr. 'CP14-1931' | CP14-1931 | | | 2 |
| *Saccharum* hybr. 'H96-3580' | UI13-00001 | 1 | | |
| *Saccharum* hybr. 'Ho06-9001' | Ho06-9001 | 8 | 6 | 6 |
| *Saccharum* hybr. 'Ho06-9002' | Ho06-9002 | 8 | 6 | 6 |
| *Saccharum* hybr. 'Ho91-552' | Ho91-552 | 1 | 6 | 6 |
| *Saccharum* hybr. 'HoCP96-540' | HoCP96-540 | 1 | 6 | 6 |
| *Saccharum* hybr. 'L09-105' | L09-105 | 8 | 6 | 6 |
| *Saccharum* hybr. 'L79-1002' | PI651501 | 8 | 6 | 6 |
| *Saccharum* hybr. 'L99-226' | L99-226 | 1 | | |
| *Saccharum* hybr. 'US 84–1058' | US 84–1058 | 2 | 6 | 6 |
| *Saccharum* hybr. 'US 87–1019' | US 87–1019 | 2 | 6 | 6 |
| *Saccharum* hybrid 'POJ 2725' × *Sorghum durra* | PI114375 | 1 | | |
| *Saccharum officinarum* 'Ho02-113' | Ho02-113 | 2 | | |
| *Saccharum officinarum* 'Ho02-144' | Ho02-144 | 2 | 6 | 6 |
| *Saccharum officinarum* 'Ho02-147' | Ho02-147 | 2 | | |
| *Saccharum robustum* 'MOL 6081' | UI13-00003 | 2 | 2 | 2 |
| *Saccharum spontaneum* 'IND 81–146' | PI504789 | 2 | | |
| *Saccharum spontaneum* 'Saudi Arabia' | PI576871 | 2 | 2 | 2 |
| *Saccharum spontaneum* 'SES 234' | PI495752 | 2 | | |
| *Saccharum arundinaceum* 'UI11-00040'† | UI11-00040 | 1 | 1 | 1 |
| *Saccharum arundinaceum* 'US 67-0009-02'† | PI318615 | 1 | 1 | 1 |
| *Saccharum arundinaceum* 'US 71-0122-01'† | PI367838 | 1 | 1 | 1 |
| (*Saccharum arundinaceum* × *Miscanthus*) 'Purple People Greeter'† | UI11-00041 | 1 | 1 | 1 |

In each year (2014–2016), plants were grown in a greenhouse that provided decreasing day length of 1 min d$^{-1}$ via supplemental light from high intensity discharge (HID) lamps starting when natural day length reached 12.5 h in Urbana, IL (14 September) until day length reached 11 h (13 December), then held constant until natural day length exceeded this value on 22 February.

†*Saccharum arundinaceum*, *arundinaceum* 'US 67-0009-02', *Saccharum arundinaceum* 'US 71-0122-01', and the interspecific hybrid (*Saccharum* × *Miscanthus*) 'Purple People Greeter' were grown in a separate greenhouse under natural day length in Urbana, IL.

greenhouse-grown stock plants (cut 15 cm above the soil surface; Table 1). Six pots of each *Miscanthus* genotype were randomly selected as controls and no additional treatments to alter flowering time were applied to these. On 5 September 2014, six pots of each *Miscanthus* genotype were cut 15 cm above the soil surface; three of these pots were left in the greenhouse to regrow (cut treatment), and the other three pots were moved to a 4 ˚C cold room for four weeks then returned to the same greenhouse to regrow (cut plus cold treatment). The cut and cut plus cold treatments were applied to a new set of *Miscanthus* pots every 4 weeks for a total of five consecutive months (i.e. through 26 December 2014). Data on flowering time was recorded weekly from 22 Aug 2014 to 30 April 2015.

The 2015–2016 greenhouse experiment (Expt. 1b) was initiated on 2–3 June 2015. For each of 15 *Saccharum* genotypes, from 1–6 pots were established via stem cuttings (Table 2). In addition to the 10 *Miscanthus* genotypes used in previous year's experiment, 13 additional *M. sacchariflorus* genotypes were included, for a total of 23 *Miscanthus* genotypes in this year's experiment (Table 1). For each of the 23 *Miscanthus* genotypes, three control pots were established from divisions of greenhouse-grown stock plants (cut 15 cm above the soil surface; Table 1). Additionally, for 15 *M. sacchariflorus* of the 23 *Miscanthus* genotypes, eight dormant divisions (quarters of pots) and bare-root rhizomes pieces (5–10 cm long, wrapped in moist paper and placed in sealed plastic bags) were stored at 4 ˚C in the previous autumn (2014) and used to establish new pots in the greenhouse in a time series during the 2015 growing season (Table 1). Every 4 weeks from 3 June to 16 September 2015, stored *Miscanthus* genotypes were planted in the greenhouse for a total of four sets (establishment time points), with two pots per genotype from stored divisions and three pots from bare-root rhizomes (1–3 rhizomes per pot) for each set. Data on flowering time was recorded weekly from 1 Aug 2015 to 30 April 2016.

The 2016–2017 experiment (Expt. 1c) was initiated on 26–29 July 2016. The 23 *Miscanthus* genotypes were the same as for the previous year's experiment (Table 1). In addition to the 15 *Saccharum* genotypes used in 2015–2016 experiment, eight new genotypes were included (Table 2). Control pots for both *Miscanthus* and *Saccharum* were prepared using the same methods as previous years' experiments. For 15 *M. sacchariflorus* of the 23 *Miscanthus* genotypes, 18 divisions (quarters of pots) were stored at 4 ˚C at the time that the control pots were established in the greenhouse (Table 1). On 6 September, an initial set of six stored divisions per *Miscanthus* genotype were removed from cold storage and three were planted in the greenhouse running the 1 min d$^{-1}$ decreasing photoperiod protocol and another three divisions were planted in another greenhouse with a constant 13 h d$^{-1}$ day length. In total, three sets of 4 ˚C *Miscanthus* divisions were planted in each greenhouse at 4-week intervals from September to November. Data on flowering time was recorded weekly from 1 October 2016 to 30 April 2017.

## Experiment 2: Effect of day length on flowering time of *M. sinensis*, *M. floridulus*, and *M.* ×*giganteus* '1993–1780' in controlled environment chambers

In total, 33 *Miscanthus* genotypes and two *Sorghum bicolor* controls (one short-day and one day-neutral) were studied (Table 3). The *Miscanthus* genotypes included 25 *M. sinensis* from known locations in China and Japan, representing latitudes ranging from 19 to 45˚ N, three ornamental *M. sinensis* cultivars, two *M. floridulus* from New Guinea and New Caledonia, two diploid *M.* ×*giganteus* (one ornamental cultivar and one natural hybrid), and the leading biomass cultivar control, the triploid *M.* ×*giganteus* '1993–1780'. The *M. sinensis* genotypes studied here represent six genetic groups that were previously identified by Clark *et al.* [23, 24]. Although detailed source location information for the four ornamental cultivars and the *M.* ×*giganteus* '1993–1780' control is not available, their *M. sinensis* ancestry was previously shown to be from the Southern Japan genetic group (Table 3 [23, 24]).

Plants were established in 7 L pots in controlled environment chambers under constant long days (15 h). After 42–61 d of establishment in the chamber, all the aboveground stems of the *Miscanthus* plants were cut to 5 cm above the soil surface and then subjected to one of three day length treatments: 15 h, 12.5 h, and 10 h. For each combination of genotype and day length treatment, three replicate pots were tested. The temperature was a constant 23 ˚C for the duration of the experiment. To each pot, 35 g of slow release fertilizer (Osmocote Pro 17-

**Table 3. The 33 *Miscanthus* genotypes and two *Sorghum* controls included in a study on the effect of day length on flowering time, conducted in controlled environment chambers.**

| Entry | Lat† | Long | Genetic group‡ | Genetic group color code | Days to first flowering | | |
|---|---|---|---|---|---|---|---|
| | | | | | 10 h | 12.5 h | 15 h |
| *M. sinensis* 'Teshio' | 44.9 | 141.9 | Northern Japan | Blue | | | 66 |
| *M. sinensis* 'EBI-2008-51c' | 43.5 | 142.7 | Northern Japan | Blue | | 42 | 67 |
| *M. sinensis* 'EBI-2008-32a' | 43.4 | 141.4 | Northern Japan | Blue | | | 83 |
| *M. sinensis* 'Tōhoku-2010-015a' | 40.2 | 140.2 | Northern Japan | Blue | | | 105 |
| *M. sinensis* 'Koike-11a' | 38.0 | 138.4 | Southern Japan | Yellow | | | 126 |
| *M. sinensis* 'Koike-12b' | 36.7 | 137.2 | Southern Japan | Yellow | | | 130 |
| *M. sinensis* 'Sugadaira' | 36.0 | 138.1 | Southern Japan | Yellow | | | 96 |
| *M. sinensis* 'Koike-21c' | 32.2 | 130.4 | Southern Japan | Yellow | 49 | 61 | 164 |
| *M. sinensis* 'Miyazaki' | 31.8 | 131.4 | Southern Japan | Yellow | 44 | 61 | 167 |
| *M. sinensis* 'Flamingo' | | | Southern Japan | Yellow | | | 121 |
| *M. sinensis* 'Gracillimus' | | | Southern Japan | Yellow | | | 194 |
| *M. sinensis* × *M. sacchariflorus* BC 'Nippon' | | | Southern Japan | Yellow | 26 | 56 | 74 |
| *M. sinensis* ssp. *condensatus* 'Cabaret' | | | Southern Japan | Yellow | | 109 | 229 |
| *M. ×giganteus* '1993–1780' | | | Southern Japan | Yellow | 98 | 71 | 140 |
| *M. sinensis* 'PMS-436' | 41.3 | 123.7 | Korea/North China | Red | | | 115 |
| *M. sinensis* 'PMS-438' | 41.3 | 123.7 | Korea/North China | Red | | | 72 |
| *M. sinensis* 'PMS-164' | 37.3 | 114.3 | Yangtze-Qinling | Green | | | 130 |
| *M. sinensis* 'PMS-161' | 35.7 | 112.3 | Yangtze-Qinling | Green | | | 133 |
| *M. sinensis* 'PMS-159' | 34.1 | 111.0 | Yangtze-Qinling | Green | | | 96 |
| *M. sinensis* 'PMS-130' | 33.5 | 105.1 | Yangtze-Qinling | Green | 42 | | 119 |
| *M. sinensis* 'PMS-204' | 31.7 | 114.9 | Yangtze-Qinling | Green | | | 170 |
| *M. sinensis* × *M. sacchariflorus* 'PMS-300' | 30.8 | 120.1 | Yangtze-Qinling | Green | | | 212 |
| *M. sinensis* 'PMS-306' | 29.9 | 118.8 | Yangtze-Qinling | Green | | 84 | 173 |
| *M. sinensis* 'PMS-314' | 26.5 | 119.6 | Yangtze-Qinling | Green | | | 166 |
| *M. sinensis* 'PMS-226' | 26.6 | 106.8 | Sichuan Basin | Orange | 56 | 76 | 189 |
| *M. sinensis* 'Onna-1a' | 26.5 | 126.8 | SE China plus tropical | Purple | | | 274 |
| *M. sinensis* 'Uruma-1b' | 26.3 | 127.9 | SE China plus tropical | Purple | | | 360 |
| *M. sinensis* 'PMS-347' | 24.2 | 115.9 | SE China plus tropical | Purple | 81 | 91 | 247 |
| *M. sinensis* 'PMS-359' | 22.9 | 112.3 | SE China plus tropical | Purple | 63 | 81 | 179 |
| *M. sinensis* 'PMS-375' | 19.6 | 110.3 | SE China plus tropical | Purple | 91 | 142 | |
| *M. sinensis* 'PMS-382' | 18.9 | 109.5 | SE China plus tropical | Purple | 91 | 184 | |
| *M. floridulus* 'NG77-022' | -3.6 | 143.6 | SE China plus tropical | Purple | | 95 | 135 |
| *M. floridulus* 'US56-0022-03' | -20.9 | 165.3 | SE China plus tropical | Purple | | 114 | |
| *S. bicolor* '100M' ($Ma_1Ma_2Ma_3Ma_4$) | | | | | 52 | 73 | 138 |
| *S. bicolor* '38M' ($ma_1ma_2ma_3^RMa_4$) | | | | | 60 | 60 | 50 |
| Average days to first flowering for 33 *Miscanthus* genotypes | | | | | 64 | 90 | 151 |

Cultivar Nippon is sold as *M. sinensis* but has been shown by Clark *et al.* [23] to be a cross between *M. sinensis* and *M. sacchariflorus* backcrossed to *M. sinensis*. All entries were diploid, except for *M. ×giganteus* '1993–1780', which is triploid.

†Empty cells indicate no data was available.

‡*M. sinensis* genetic groups determined from Clark *et al.* [23, 24]. For interspecific hybrids between *M. sacchariflorus* and *M. sinensis*, the dominant *M. sinensis* genetic group is shown.

5-11, 6 months; ICL Specialty Fertilizers, Dublin, OH, USA) was added at planting and after 6 months. Drip irrigation was provided to each pot.

Data were recorded on the number of days to first flagging and first flowering. At the end of the experiment, data were taken on number of total culms and number of reproductive shoots, number of leaves per culm (~number of nodes), and culm length. An additional trait, reproductive shoot ratio, was obtained by dividing number of reproductive shoots over the total culm count. Thus, a total of seven traits were studied. The experiments were ended after at least 80 d with no change in flowering, which was at least 188 d from cutting for the 10 h and 12.5 h treatments and 352 d for the 15 h treatment.

## Statistical analysis

For Experiment 1, analyses of variance (ANOVAs) were conducted to assess the effects on *Miscanthus* flowering time of the treatments performed in each year. For the 2014–2015 experiment, the treatments included cut, and cut plus cold performed monthly from September to January and controls. For the 2015–2016 experiment, the treatments were plantings of pot divisions or rhizomes from cold storage, performed monthly from June to September, and controls. For the 2016–2017 experiment, the treatments were plantings of cold storage pot divisions from September to November, grown under two different day lengths, and controls. ANOVAs were conducted with SAS Procedure MIXED (SAS Institute Inc., Cary, NC, USA) for each year's experiment based on the subset of *Miscanthus* genotypes that flowered following the model:

$$Y_{ijkl} = \mu + T_i + G_j + M_k + TG_{ij} + GM_{jk} + TM_{ik} + TGM_{ijk} + R_l + \varepsilon_{ijkl},$$

where $Y$ is first flowering time, $T$ represents treatment, $G$ equals genotype, $M$ represents month, $R$ represents replication, and $TG$, $GM$, $TM$, $TGM$ represent respective interactions of aforementioned model terms, and $\varepsilon$ is error. Treatment, genotype and month were set as fixed and replication was set as random. To better evaluate flowering time diversity between and within *Miscanthus* and *Saccharum*, ANOVAs were also conducted in SAS Procedure MIXED to test the effects on flowering-time of genus (*Miscanthus*, *Saccharum*), and genotype nested within genus as fixed effects, for the subset of genotypes that flowered; for *Miscanthus*, only the control pots were included in this analysis. Weekly flowering data were plotted in R [25] for visualization. Association between the latitude of origin for the *Miscanthus* genotypes and flowering time was also evaluated by linear regression using R *lm* function [25].

For Experiment 2, ANOVAs were conducted with SAS Procedure MIXED to assess the fixed effects of genotype, day length (10 h, 12.5 h, and 15 h) and their interactions on flowering traits (days to first flagging and first flowering) and morphological traits (culm length, number of leaves per culm, number of total culms, number of reproductive shoots and reproductive shoot ratio). Tukey's HSD test (α = 0.05) was estimated to investigate differences among three day lengths for each trait. The relationships between location of origin (*i.e.* collection site), the genetic groups to which the genotypes belong, and the phenotypic traits observed in the controlled-environment chambers under three day lengths were visualized using R package *ggmap* [26] by plotting on a geographical map the location of each genotype, color coded by its previously ascertained *M. sinensis* genetic group [23, 24], along with the associated phenotypic data from this study (as bar charts with standard errors). Associations between the latitude of origin and phenotype at the different day lengths were also evaluated by linear regression using R *lm* function [25]. R codes used in figure visualization are available at https://github.com/hxdong-genetics/Geographic-map-in-Miscanthus-flowering-study.

## Results

### Experiment 1: Flowering time management of Miscanthus and sugarcane in a greenhouse

**Key findings over the three years.** Large and highly significant differences in flowering time were observed between *Saccharum* and *Miscanthus*, and among genotypes within each genus (Fig 1; Table 4). As expected *Saccharum* genotypes typically flowered later than *Miscanthus* genotypes. However, some *Saccharum* and *Miscanthus* genotypes overlapped in flowering time each year the experiment was conducted (Fig 1). Each year, the experiment was initiated ~2 months later in the season than the prior year (Expt. 1a, 25 March to 21 April 2014; Expt. 1b, 2–3 June 2015; and Expt. 1c, 26–29 July 2016) and this appeared to have had a large effect on which genotypes in each genus flowered, and it also affected the timing of flowering for the *Saccharum* genotypes (Fig 1). Early planting promoted flowering in both genera and early flowering in *Saccharum*. Over the three years, *Saccharum* genotypes were observed to flower from October to April, with flowering obtained for 13/23 genotypes in 2014–2015, 5/15 in 2015–2016, and 7/23 in 2016–2017 (Fig 1, S1–S3 Tables). For *Miscanthus* genotypes, flowering of the control pots was observed from August to April, with flowering obtained for 10/10 genotypes in 2014–2015, 22/23 in 2015–2016, and only 8/23 in 2016–2017 (Fig 1, S1–S3 Tables). In each year, there was a strong negative correlation between flowering time of the *Miscanthus* genotypes and their latitude of origin ($r^2$ = 0.89–0.90, $p$ < 0.001; Fig 2). Thus, under the short days provided, *Miscanthus* genotypes that originated from low latitudes were primarily the ones that overlapped in flowering time with *Saccharum* genotypes (Figs 1 and 2).

Some *Miscanthus* and *Saccharum* genotypes flowered consistently over the three years that the experiment was conducted, irrespective of the differences in initial planting date. Four sugarcane genotypes ('US84-1058', 'L09-105', 'Ho06-9001', 'Ho06-9002') and the intergeneric hybrid (*S. arundinaceum* × *Miscanthus*) 'Purple People Greeter' flowered during each of the three years that Expt. 1 was conducted (Fig 1). Two additional sugarcane genotypes, 'L79-1002' and 'Ho91-552' flowered in two out of the three years. For *Miscanthus*, control pots for eight of the 10 genotypes tested in the 2014–2015 experiment also flowered in 2015–2016 experiment. However, of the 23 *Miscanthus* genotypes tested in both the 2015–2016 and 2016–2017 experiments, only eight genotypes had control pots that flowered in both years (Fig 1, S2 and S3 Tables).

**Experiment 1a (2014–2015).** In the 2014–2015 greenhouse experiment, more than half of the tested *Saccharum* genotypes flowered, and this was a substantially larger percentage than that observed in the subsequent years' experiments in which the stem cuttings were planted later. Moreover, the seven *Saccharum* genotypes that flowered in multiple years flowered earliest in the 2014–2015 experiment. Four of the *Saccharum* genotypes flowered twice during the 2014–2015 experiment, once in the late autumn or early winter and a second time in mid-winter or spring (Fig 1). In contrast, none of the *Saccharum* genotypes flowered twice in the subsequent experiments. The first flowering flush was observed from October 2014 to December 2015, with *S. spontaneum* 'Saudi Arabia' being the first to flower on 3 October 2014 and *S.* hybr. 'HoCP96-540' being the last on 13 December 2014 (Fig 1, S1 Table). One *Saccharum* hybrid, 'Ho91-552', flowered a second time in January 2015 and three *Saccharum* hybrids, 'L09-105', 'L79-1002' and 'Ho06-9002', had second flush of flowering in April 2015 (Fig 1).

For *Miscanthus*, the control pots of the 2014–2015 experiment flowered only from August through December (Fig 1, S1 Table). The earliest flowering genotype was the northernmost *M. sinensis* 'PMS-436' (41.3° N; first flowering date: 20 August 2014), and the latest flowering genotype was the southernmost *M. sinensis* 'PMS-375' (19.6° N; first flowering date: 27 November 2014). Notably, the cut treatment and the cut plus cold treatment extended the

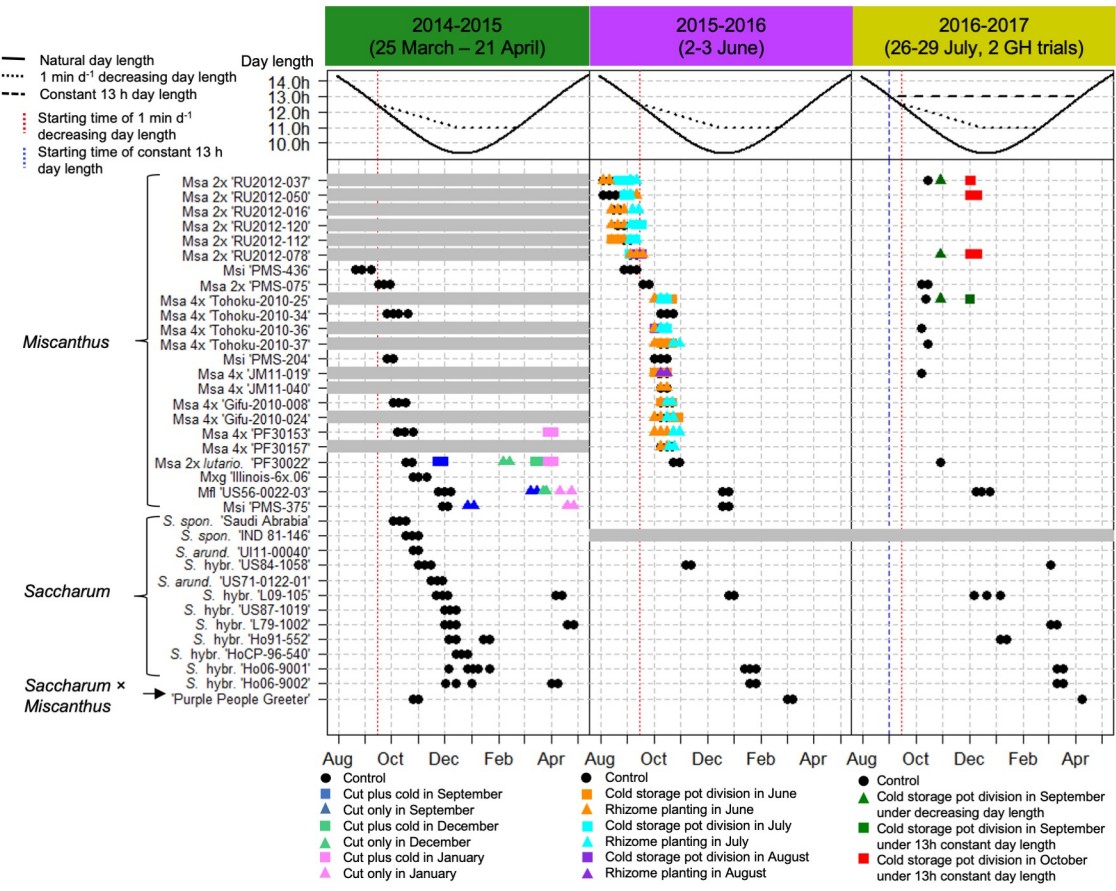

**Fig 1.** *Miscanthus* and *Saccharum* **flowering time in a series of greenhouse experiments over three years.** In each year (2014–2016), plants were grown in a greenhouse that provided decreasing day length of 1 min d$^{-1}$ via supplemental light from high intensity discharge (HID) lamps starting when natural day length reached 12.5 h in Urbana, IL (14 September; red vertical dashed line) until day length reached 11 h (13 December), then held constant until natural day length exceeded this value on 22 February. In 2016, an additional set of *Miscanthus* plants were also grown in a second greenhouse at Urbana, IL, in which day length was held at a constant 13 h via supplemental HID lamps, starting on 2 September until natural day length exceeded this value on 9 April. The combinations of symbols and colors represent additional cultural treatments applied to *Miscanthus* pots, as shown in the legend. In 2014 pots of *Miscanthus* and *Saccharum* were established between 25 March to April 21; *Miscanthus* treatments included 1) cutting plants ~15 cm above the soil in September, December and January and allowing them to immediately regrow, 2) cutting the plants and storing them for 1 mo at 4 °C before returning them to the greenhouse to regrow, and 3) uncut controls. In 2015 all *Saccharum* pots were established on 2–3 June; *Miscanthus* treatments were 1) stored divisions (planted every 4 wks starting on 3 June 2015), 2) rhizomes (planted every 4 wks starting on 3 June 2015), and 3) controls (actively growing plants cut ~15 cm above the soil surface on 3 June). The 2016 experiment was initiated on 26–29 July; control pots of *Miscanthus* cut ~15 cm above the soil surface were compared with a set of pots stored at 4 °C and returned at 4-wk intervals from September to November to one greenhouse with 1 min d$^{-1}$ decreasing photoperiod and to another greenhouse with a constant 13 h d$^{-1}$ day length. Only genotypes that flowered in at least one of the experiments are shown. Grey shaded lines indicate that plant materials were not included in that year's experiment. Over the three years, 23 *Miscanthus* genotypes including *M. sinensis* (Msi), *M. sacchariflorus* (Msa), *M. ×giganteus* (M×g), and *M. floridulus* (Mfl) flowered, and a total of 12 *Saccharum* accessions including nine commercial sugarcanes (*S.* hybr.), and two *S. spontaneum* (*S. spon.*) flowered. *Saccharum arundinaceum* (*S. arund.*) 'UI11-00040', 'US 71-0122-01', and the interspecific hybrid (*Saccharum* × *Miscanthus*) 'Purple People Greeter' also flowered, though these were grown in a separate greenhouse under natural day length. Flowering time was recorded weekly from August to April.

flowering time into the late winter and spring for four of the *Miscanthus* genotypes (*M. sacchariflorus* 4x 'PF30153', *M. sacchariflorus* ssp. *lutarioriparius* 'PF30022', *M. floridulus* 'US56-002-03', and *M. sinensis* 'PMS-375'). The treatments in September, December, and January resulted in *Miscanthus* plants that flowered, but the treatments in October and November did not produce any flowering plants (Fig 1, S1 Table). ANOVA indicated that genotype, treatment, month of treatment application, and interactions all had significant effects on days to

**Table 4. Effect of genus (*Miscanthus* and *Saccharum*) and genotype within each genus on days to first flower for a series of experiments conducted in a greenhouse at Urbana, IL over three years.**

| Experiment | Model Term | DF | Mean Squares | F value | Pr(>F) |
|---|---|---|---|---|---|
| 2014–2015 | Genus | 1 | 61910.0 | 2074.7 | <0.001 |
| | Genotype within in genus | 20 | 8929.1 | 299.2 | <0.001 |
| | *Miscanthus* | 9 | 18330.7 | 776.1 | <0.001 |
| | *Saccharum* | 11 | 1237.0 | 11.8 | <0.001 |
| | Residuals | 221 | 29.8 | | |
| 2015–2016 | Genus | 1 | 87096.2 | 30821.6 | <0.001 |
| | Genotype within in genus | 22 | 4197.9 | 964.3 | <0.001 |
| | *Miscanthus* | 20 | 3889.1 | 1926.2 | <0.001 |
| | *Saccharum* | 2 | 7286.7 | 743.5 | <0.001 |
| | Residuals | 50 | 1.4 | | |
| 2016–2017 | Genus | 1 | 74127.2 | 1820.7 | <0.001 |
| | Genotype within in genus | 11 | 2131.6 | 52.4 | <0.001 |
| | *Miscanthus* | 7 | 1201.7 | 367.9 | <0.001 |
| | *Saccharum* | 4 | 3759.0 | 36.5 | <0.001 |
| | Residuals | 16 | 40.7 | | |

Only entries that flowered in each year were included in ANOVA analyses. Note that the 'Genotype within genus' term in ANOVA table could be fractioned into two sub-terms '*Miscanthus*' and '*Saccharum*', which were also tested separately.

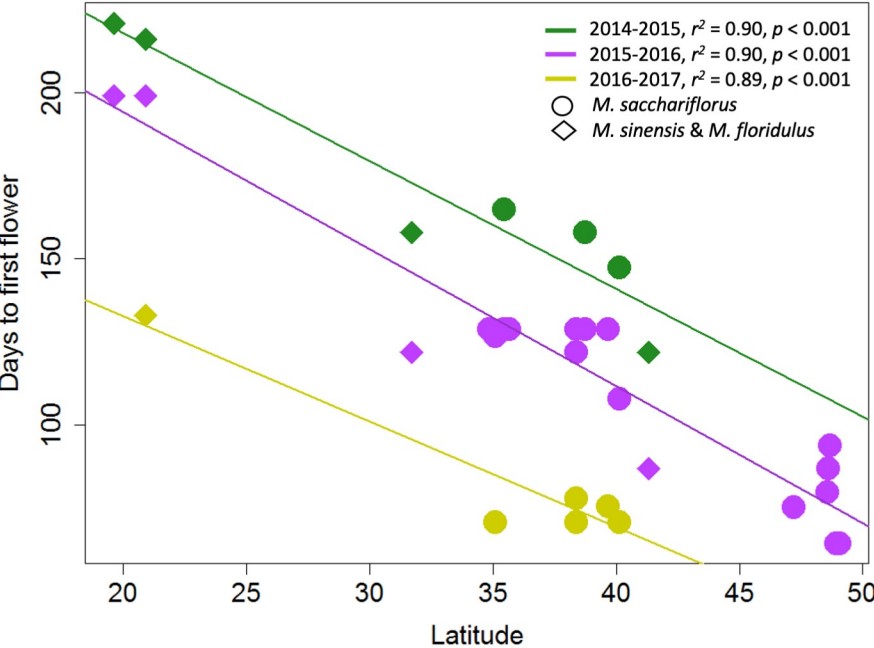

**Fig 2. Relationships between absolute value of latitude at collection site (x-axis) and date of first flowering (y-axis) for *Miscanthus sacchariflorus* (circles) and *M. sinensis* and *M. floridulus* (diamonds) genotypes when grown in a greenhouse at Urbana, IL that provided decreasing day length of 1 min d$^{-1}$ via supplemental light from high intensity discharge (HID) lamps starting when natural day length reached 12.5 h (14 September).** Experiments were conducted in three consecutive years: 2014–2015 (green), 2015–2016 (purple), and 2016–2017 (yellow).

first flowering (Table 5). Among the four genotypes that flowered after treatments, two tropical genotypes, *M. floridulus* 'US56-0022-03' (20.9° S) and *M. sinensis* 'PMS-375' (19.6° N), flowered only after the cut treatment rather than the cut plus cold treatment, whereas the other two genotypes *M. sacchariflorus* 4x 'PF30153' and *M. sacchariflorus* ssp. *lutarioriparius* 'PF30022' flowered after the cut plus cold treatment only or under both treatments. The January cut plus cold treatment for *M. sacchariflorus* 4x 'PF30153' and *M. sacchariflorus* ssp. *lutarioriparius* 'PF30022', and the January cut treatment for *M. floridulus* 'US56-0022-03' and *M. sinensis* 'PMS-375' resulted in plants that flowered in April 2015, which overlapped with the second flowering of *Saccharum* hybrids, 'L09-105', 'L79-1002' and 'Ho06-9002' (Fig 1).

**Experiment 1b (2015–2016).** Four *Saccharum* genotypes, including 'US84-1058', 'L09-105', 'Ho06-9001', and 'Ho06-9002', flowered from November 2015 to January 2016 (Fig 1, S2 Table). The intergeneric hybrid (*S. arundinaceum × Miscanthus*) 'Purple People Greeter' also flowered in early March. For *Miscanthus*, 22 of the 23 genotypes flowered. Flowering time of the *Miscanthus* controls ranged from 5 August 2015 to 19 December 2015. The earliest *Miscanthus* genotypes were *M. sacchariflorus* from eastern Russia (47.2–49.1° N), including 'RU2012-037', 'RU2012-050', 'RU2012-016', 'RU2012-120', and 'RU2012-112', which flowered in August 2015. In contrast, the two southernmost genotypes, *M. floridulus* 'US56-002-03' and *M. sinensis* 'PMS-375' flowered latest in mid-December, similar to that observed in the 2014–2015 experiment. Thus, the *Miscanthus* and *Saccharum* genotypes that were best synchronized in flowering time were *M. floridulus* 'US56-0022-03', *M. sinensis* 'PMS-375' and *S.* hybr. 'L09-105', which all flowered from mid- to late December (Fig 1, S2 Table).

*Miscanthus* pot divisions and rhizomes that were stored at 4 °C then planted in the greenhouse during June or July flowered in high frequency, but few or no genotypes flowered when cold-stored materials were planted in August or September, again demonstrating that date of establishment had a large effect on presence or absence of flowering (Fig 1, S2 Table). However, flowering time of the cold-stored *Miscanthus* divisions and rhizomes was similar to that of the controls. ANOVAs indicated that all tested model terms had significant effects except for treatment by month interaction and genotype by treatment by month interaction (Table 5). Of the 15 *M. sacchariflorus* genotypes included in the treatments, 11 flowered from stored pot divisions (seven each from June and July plantings but only one from August and zero from September; S2 Table), and all flowered when pots were newly established from rhizomes (15 from June, 12 from July, one from August, and zero from September; S2 Table).

**Experiment 1c (2016–2017).** Six *Saccharum* genotypes, including 'L09-105', 'Ho91-552', 'US84-1058', 'Ho06-9001', 'Ho06-9002', and 'L79-1002' flowered from December 2016 to March 2017, though with a gap from mid-January through all of February (Fig 1, S3 Table). In addition, the intergeneric hybrid (*S. arundinaceum × Miscanthus*) 'Purple People Greeter' also flowered in early April. The *Saccharum* genotypes that flowered in the 2016–2017 experiment included all of the genotypes that flowered in 2015–2016 plus two ('L79-1002' and 'Ho91-552'), but in the 2016–2017 experiment, they flowered later in the season, consistent with the later planting of this trial.

For the *Miscanthus*, only 10 of the 23 genotypes flowered, and of these, two flowered only after cold-stored divisions were planted in September or October (Fig 1, S3 Table). However, of the 15 *Miscanthus* genotypes included in the cold storage treatments, only four flowered (Fig 1, S3 Table). An ANOVA of just the four entries that flowered to evaluate effects of genotype, two day length treatments, month and their interactions on days to first flowering, detected significant effects of genotype and day length (Table 5). The September planting of three *M. sacchariflorus* genotypes, 'RU2012-037', 'RU2012-078', and 'Tohoku-2010-025', flowered at the end of October 2016 under the 1 min d$^{-1}$ decreasing length. Under the 13 h constant day length, the September planting of 'Tohoku-2010-025' and the October planting of

**Table 5. Effects of treatments on days to first flowering for *Miscanthus* in a series of greenhouse experiments conducted in Urbana, IL over three years.**

| Experiment | Model Term | DF | Mean squares | F value | Pr(>F) |
|---|---|---|---|---|---|
| 2014–2015 | Genotype | 3 | 7529.4 | 2384.3 | <0.001 |
| | Treatment | 2 | 47481.0 | 15035.2 | <0.001 |
| | Month | 2 | 9972.3 | 3157.8 | <0.001 |
| | Genotype × Treatment | 3 | 760.5 | 240.8 | <0.001 |
| | Genotype × Month | 3 | 1558.5 | 493.5 | <0.001 |
| | Treatment × Month | | | | |
| | Genotype × Treatment × Month | | | | |
| | Residuals | 23 | 3.2 | | |
| 2015–2016 | Genotype | 14 | 3418.3 | 749.8 | <0.001 |
| | Treatment | 2 | 15.2 | 3.3 | 0.042 |
| | Month | 3 | 387.8 | 85.1 | <0.001 |
| | Genotype × Treatment | 22 | 76.1 | 16.7 | <0.001 |
| | Genotype × Month | 11 | 136.8 | 30.0 | <0.001 |
| | Treatment × Month | 1 | 7.7 | 1.7 | 0.197 |
| | Genotype × Treatment × Month | 2 | 7.4 | 1.6 | 0.206 |
| | Residuals | 68 | 4.6 | | |
| 2016–2017 | Genotype | 3 | 959.6 | 381.9 | <0.001 |
| | Treatment | 2 | 3623.7 | 1442.1 | <0.001 |
| | Month | 1 | 1.3 | 0.5 | 0.493 |
| | Genotype × Treatment | 2 | 1.4 | 0.5 | 0.594 |
| | Genotype × Month | | | | |
| | Treatment × Month | | | | |
| | Genotype × Treatment × Month | | | | |
| | Residuals | 13 | 2.5 | | |

In each year (2014–2016), plants were grown in a greenhouse that provided decreasing day length of 1 min $d^{-1}$ via supplemental light from high intensity discharge (HID) lamps starting when natural day length reached 12.5 h in Urbana, IL (14 September) until day length reached 11 h (13 December), then held constant until natural day length exceeded this value on 22 February. In 2016, an additional set of *Miscanthus* plants were also grown in a second greenhouse at Urbana, IL, in which day length was held at a constant 13 h via supplemental HID lamps, starting on 2 September until natural day length exceeded this value on 9 April. In the 2014–2015 experiment, treatments included 1) cutting plants ~15 cm above the soil in September, December and January and allowing them to immediately regrow, 2) cutting the plants and storing them for 1 month at 4 ˚C before returning them to the greenhouse to regrow, and 3) uncut controls. In the 2015–2016 experiment, treatments were stored rhizomes, divisions of stored pots (each planted every 4 weeks starting on 3 June 2015), and controls (actively growing plants cut ~15 cm above the soil surface on 3 June). In the 2016–2017 experiment, treatments included control pots cut ~15 cm above the soil surface that were compared with a sets of pots stored at 4 ˚C and returned at 4-week intervals from September to November to one greenhouse with 1 min $d^{-1}$ decreasing photoperiod and to another greenhouse with a constant 13 h $d^{-1}$ day length. Only genotypes that flowered were included in ANOVA analyses. Except for the residual term, empty cells indicate that model terms could not be tested due to lack of data.

'RU2012-037', 'RU2012-050', and 'RU2012-078' flowered in early December 2016. None of the November plantings of cold-stored divisions flowered. Thus, the control pots of *M. floridulus* 'US56-0022-03' and the October plantings of *M. sacchariflorus* 'RU2012-050' and 'RU2012-078' synchronized in flowering time with *S.* hybr. 'L09-105' during early December 2016.

## Experiment 2: Effect of day length on flowering time of *M. sinensis*, *M. floridulus*, and *M. ×giganteus* '1993–1780' in controlled environment chambers

ANOVAs indicated that genotype, day length, and genotype by day length interactions had significant effects on each of the seven flowering and morphological traits (Table 6). All 35

entries (including 33 *Miscanthus* and two *S. bicolor* controls) flowered under one or more of the tested day lengths (10, 12.5, and 15 h). However, only five mostly subtropical *M. sinensis* genotypes ('Koike-21c', 32.2˚ N; 'Miyazaki', 31.8˚ N; 'PMS-226', 26.6˚ N; 'PMS-347', 24.2˚ N; 'PMS-359', 22.9˚ N), one ornamental cultivar ('Nippon'), and the biomass control M×g '1993–1780' flowered under each of the tested day lengths, and these genotypes behaved similarly to the short-day *S. bicolor* control '100M' (Ma$_1$Ma$_2$Ma$_3$Ma$_4$; [27, 28]), with flowering earliest at 10 h, intermediate at 12.5 h, and latest at 15 h (Fig 3, Table 3). Similarly, for the *Miscanthus* genotypes that flowered under 10 h and 12.5 h, average days to first flower (64 and 90 d, respectively; Table 3) were earlier than those that flowered at 15 h (151 d), though the difference between 10 h and 12.5 h was not significant at α = 0.05 based on Tukey's HSD test (Fig 3). The day-neutral *S. bicolor* control '38M' (ma$_1$ma$_2$ma$_3$$^R$Ma$_4$; [27, 28]) flowered quickly and at about the same time regardless of day length (50 to 60 days after cutting), as expected; however, none of the *Miscanthus* genotypes behaved similarly (Fig 3, Table 3).

Of the 33 *Miscanthus* genotypes, all but three tropical accessions flowered under the 15 h day length (Fig 3, Table 3), and the highest ratio of reproductive shoots to total number of culms was typically observed for 15 h days (Fig 4B, S4 Table). With the 15 h day length, days to first flower for the *M. sinensis* genotypes ranged from 66 d to 360 d (Table 3). However, of the five *Miscanthus* genotypes ('PMS-359', 'PMS-375', 'PMS-382', 'NG77-022', 'US56-0022-03') that originated from the tropics (23.5˚ S to 23.5˚ N), only two flowered under 15 h days, but each flowered under 12.5 h days, and one (*M. floridulus* 'US56-0022-03', 20.9˚ S) flowered only under 12.5 h days (Fig 3, Table 3). Similarly, for four of the five tropical *Miscanthus* genotypes, reproductive shoot ratio was highest under 12.5 h days, in contrast to those that originated at higher latitudes (Fig 4B, S4 Table).

At 10 h day length, there was a strong negative correlation between the latitude of origin and days to first flower ($r^2$ = 0.88), but at 12.5 and 15 h, the correlations were only moderately negative (Fig 3). However, none of the 12 *M. sinensis* genotypes that originated from latitudes exceeding 34˚ N flowered under 10 h days, and only one ('EBI-2008-051c') of these flowered under 12.5 h days, yet all flowered under 15 h days (Fig 3, Table 3). Notably, six of these northern (i.e. temperate) *M. sinensis* genotypes flagged under 10 h and/or 12.5 h day lengths but did not proceed to flower (Fig 3; 'PMS-130', 'PMS-159', 'PMS-161', 'PMS-438', 'Tohoku-2010-015a', and 'Koike-11a'). Some subtropical *M. sinensis* genotypes also only flowered under 15 h days (e.g. 'PMS-314', 'Onna-1a', and 'Uruma-1b'), yet others flowered under 12.5 and 15 h days or all three tested day lengths, indicating that the subtropics is a transition zone with a mixture of day length response types (Fig 3). Moreover, in addition to not flowering under short days, the northern *M. sinensis* genotypes responded to 10 and 12.5 h days by producing very short culms, with the shortest days resulting in the shortest culms (Figs 5 and 6, S4 Table).

Culm length of the *M. sinensis* and *M. floridulus* genotypes was strongly and negatively correlated with latitude of origin under 10 h days ($r^2$ = 0.81) and 12.5 h days ($r^2$ = 0.63), but the relationship was weak under 15 h days ($r^2$ = 0.09; Fig 5A). Among all 33 *Miscanthus* genotypes, Tukey's HSD test (α = 0.05) indicated that culm length was significantly different across three day length treatments. Nearly all the *Miscanthus* entries achieved maximal culm length under the 15 h treatment (including the biomass cultivar M×g '1993–1780'), but the nearer to the equator an accession originated, the less of a difference in culm length between the short day treatments and the 15 h day treatment (Fig 5A). For example, *M. floridulus* 'NG77-022' from 3.6˚ S produced similarly long culms under all three day lengths (Fig 5A and S1 Fig, S4 Table). Two tropical genotypes ('PMS-382' and 'US56-0022-03'), two subtropical genotypes ('PMS-226' and 'Miyazaki') and one ornamental cultivar ('Cabaret') were tallest under 12.5 h days (Fig 5A, S4 Table).

**Table 6. Effects of genotype, photoperiod, and their interactions on nine flowering and morphological traits of *Miscanthus*.**

| Trait | Term | DF | Mean squares | F | Pr(>F) |
|---|---|---|---|---|---|
| Days to first flagging | Genotype | 32 | 10833.5 | 66.2 | <0.001 |
| | Photoperiod | 2 | 71082.7 | 434.4 | <0.001 |
| | Genotype × Photoperiod | 19 | 1540.3 | 9.4 | <0.001 |
| | Residuals | 86 | 163.6 | | |
| Days to first flowering | Genotype | 32 | 8838.1 | 46.8 | <0.001 |
| | Photoperiod | 2 | 84211.7 | 445.9 | <0.001 |
| | Genotype × Photoperiod | 19 | 1759.7 | 9.3 | <0.001 |
| | Residuals | 84 | 188.9 | | |
| Culm length | Genotype | 32 | 18186.2 | 43.1 | <0.001 |
| | Photoperiod | 2 | 229606.3 | 544.3 | <0.001 |
| | Genotype × Photoperiod | 64 | 4530.7 | 10.7 | <0.001 |
| | Residuals | 190 | 421.9 | | |
| Leaf number per culm | Genotype | 32 | 140.4 | 13.7 | <0.001 |
| | Photoperiod | 2 | 112.7 | 11.0 | <0.001 |
| | Genotype × Photoperiod | 64 | 26.4 | 2.6 | <0.001 |
| | Residuals | 190 | 10.3 | | |
| Number of reproductive shoots | Genotype | 32 | 287.4 | 18.8 | <0.001 |
| | Photoperiod | 2 | 3711.7 | 242.4 | <0.001 |
| | Genotype × Photoperiod | 64 | 243.8 | 15.9 | <0.001 |
| | Residuals | 190 | 15.3 | | |
| Total number of culms | Genotype | 32 | 16168.4 | 29.2 | <0.001 |
| | Photoperiod | 2 | 82253.1 | 148.7 | <0.001 |
| | Genotype × Photoperiod | 64 | 3746.4 | 6.8 | <0.001 |
| | Residuals | 190 | 553.2 | | |
| Reproductive shoot ratio | Genotype | 32 | 0.0 | 5.7 | <0.001 |
| | Photoperiod | 2 | 2.3 | 354.8 | <0.001 |
| | Genotype × Photoperiod | 64 | 0.1 | 10.4 | <0.001 |
| | Residuals | 190 | 0.0 | | |

Data were collected on 33 *Miscanthus* genotypes evaluated under three photoperiods (10 h, 12.5 h, 15 h) in controlled environment chambers.

*M. sinensis* genotypes that originated from high latitudes in Japan had greater numbers of leaves at 15 h than at 10 h day lengths (Fig 5B, S4 Table). In contrast, *M. sinensis* genotypes that originated from high latitudes on mainland Asia (Korea/North China and Yangtze-Qinling genetic groups) had the same or greater numbers of leaves at 10 h in comparison to 15 h (Fig 5B, S4 Table). Thus, for the Japanese accessions, the short culms observed for high-latitude accessions of *M. sinensis* under short days was achieved substantially by greater phyllochron under short days than under long days, whereas for the mainland accessions, short culms were obtained primarily via short internodes rather than by more days needed to develop a leaf. Like the northern Japanese *M. sinensis* genotypes, most of the subtropical and tropical accessions of *M. sinensis* produced more leaves under long days than under short days. However, some accessions produced similar numbers of leaves under all three day lengths tested (e.g. 'PMS-306', 29.9˚ N), and other entries, such as *M. floridulus* 'NG77-022' (3.6˚ S) and the biomass control cultivar M×g '1993–1780' produced more leaves under shorter days than longer days (Fig 5B, S4 Table).

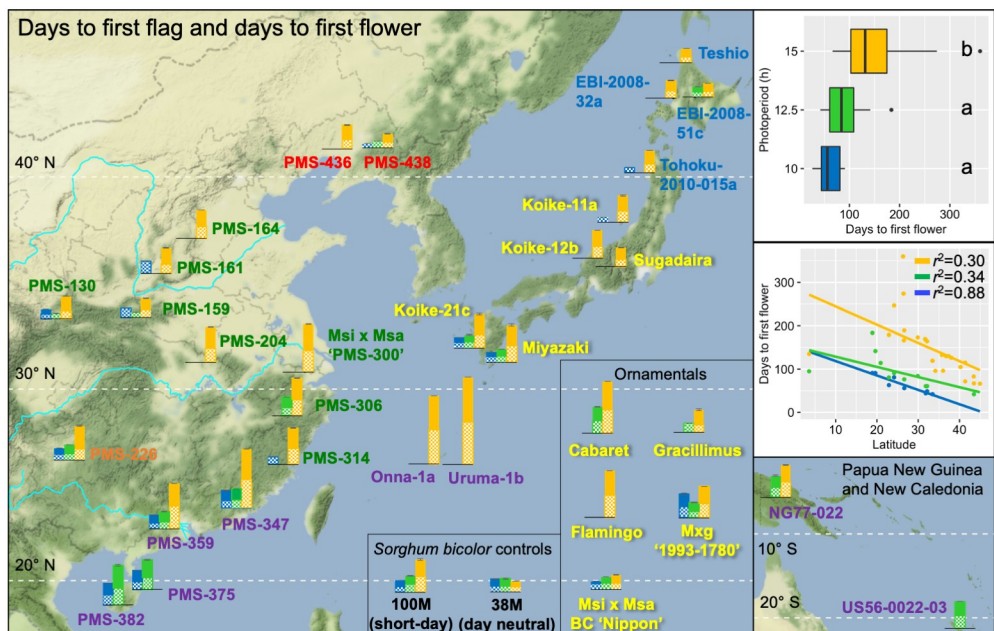

**Fig 3. Effects of day length on days to first flag and days to first flower for 33 *Miscanthus* and two *Sorghum bicolor* genotypes grown in controlled environment chambers at constant 23 °C.** The *Miscanthus* genotypes included 28 *M. sinensis*, 2 *M. floridulus*, 2 diploid *M. ×giganteus*, and 1 triploid *M. ×giganteus*. The genotypes were evaluated for response to three day-length treatments: 15 h (orange data), 12.5 h (green data) and 10 h (blue data), respectively. Pattern-filled bars represent days to first flag leaf, and solid-filled bars represent days to first flowering. Note that some *Miscanthus* genotypes flagged but did not flower. Collection sites of the wild-collected genotypes are shown by their placement on the geographic map. *Miscanthus* genotype names are printed in colors representing six *M. sinensis* genetic groups identified by Clark *et al.* [23, 24], which included Korea/North China (red), Yangtze-Qinling (green), Northern Japan (blue), Southern Japan (yellow), Sichuan Basin (orange), and Southeastern China plus tropical (purple); for interspecific hybrids between *M. sacchariflorus* and *M. sinensis*, the dominant *M. sinensis* genetic group is shown. The inset boxplots depict variation among and within the three day-length treatments; treatments labeled with the same letter were not significantly different based on Tukey's HSD test at α = 0.05. The inset regression plots show linear relationships between traits and absolute values of latitude at collection sites for the 28 *Miscanthus* genotypes with geographical information. Note that short days typically advanced flowering up to some optimum, which differed for accessions from different latitudes of origin; higher latitude accessions failed to flower under 10 and 12.5 h, whereas some low latitude accessions failed to flower under 15 h day length. Some *M. sinensis* accessions from between 20 to 25 °N (PMS-226, PMS-359, and PMS-347) responded similarly to the three tested day lengths as the *Sorghum bicolor* short-day control (100M) but most *Miscanthus* accessions responded differently in part; all of the *Miscanthus* accessions responded differently than the *S. bicolor* day-neutral control (38M).

Total number of culms for most of the *Miscanthus* genotypes was ~3–13 fold greater under 10 h than 15 h days, with intermediate numbers of culms typically resulting from 12.5 h days (Fig 4A, S4 Table). However, the two tropical *M. floridulus* ('NG77-022' and 'US56-0022-03'), four *M. sinensis* ('Flamingo', 'Koike-21c', 'Miyazaki', and 'Tohoku-2010-015a'), and the biomass control M×g '1993–1780' produced the greatest number of culms at 12.5 h. Thus, under 10 and 12.5 h day lengths, most of the *M. sinensis* genotypes from low latitudes produced a large number of tall culms, many of which flowered, whereas genotypes from high latitudes produced a large number of short culms that did not flower (Figs 4–6, S1 and S2 Figs).

Raw data for experiment 1 and experiment 2 can be found in S5 and S6 Tables, respectively.

## Discussion

### Flowering sugarcane at 40° N

Flowering was accomplished for more than half of the sugarcane genotypes in this study, in central Illinois, by growing the plants in a warm greenhouse and providing a declining

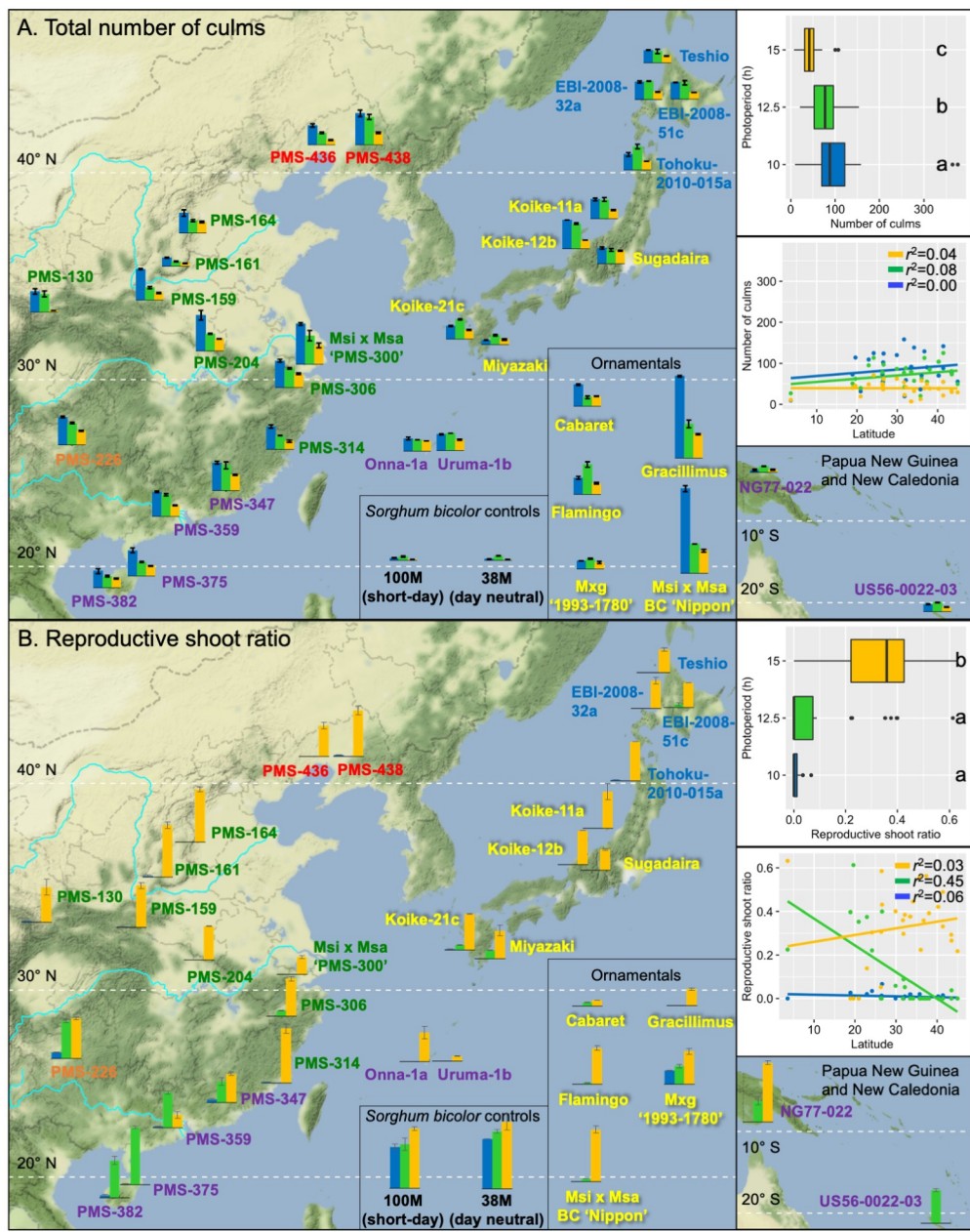

**Fig 4. Effects of day length on total number of culms (A), and reproductive shoot ratio (B) for 33 *Miscanthus* and two *Sorghum bicolor* genotypes grown in controlled environment chambers at constant 23 ˚C.** The *Miscanthus* genotypes included 28 *M. sinensis*, 2 *M. floridulus*, 2 diploid *M. ×giganteus*, and 1 triploid *M. ×giganteus*. The genotypes were evaluated for response to three day-length treatments: 15 h (orange data), 12.5 h (green data) and 10 h (blue data), respectively. Collection sites of the wild-collected genotypes are shown by their placement on the geographic map. *Miscanthus* genotype names are printed in colors representing six *M. sinensis* genetic groups identified by Clark *et al*. [23, 24], which included Korea/North China (red), Yangtze-Qinling (green), Northern Japan (blue), Southern Japan (yellow), Sichuan Basin (orange), and Southeastern China plus tropical (purple); for interspecific hybrids between *M. sacchariflorus* and *M. sinensis*, the dominant *M. sinensis* genetic group is shown. The inset boxplots depict variation among and within the three day-length treatments; treatments labeled with the same letter were not significantly different based on Tukey's HSD test at α = 0.05. The inset regression plots show linear relationships between traits and absolute values of latitude at collection sites for the 28 *Miscanthus* genotypes with geographical information.

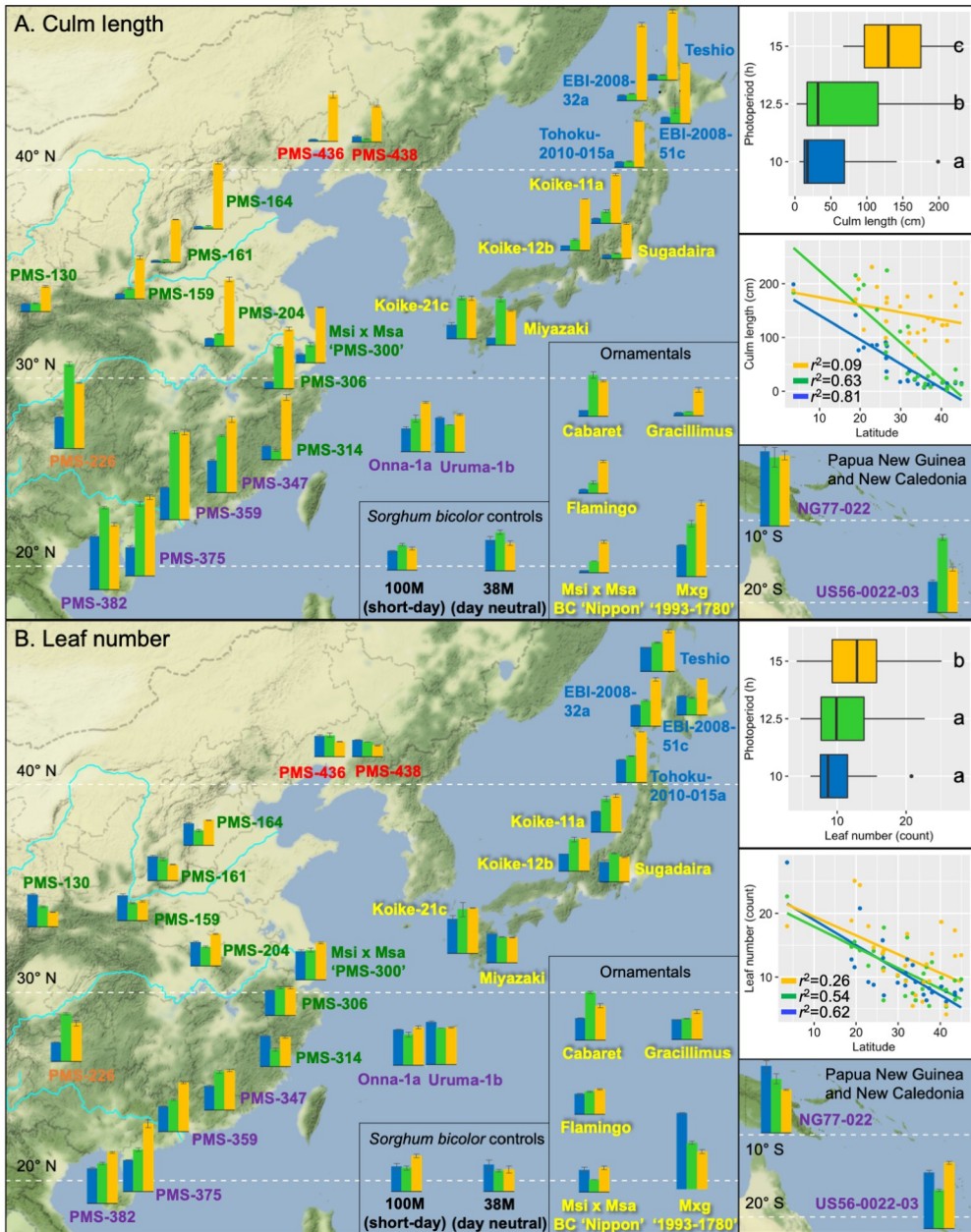

**Fig 5. Effects of day length on culm length (A), and leaf number (B) for 33 *Miscanthus* and two *Sorghum bicolor* genotypes grown in controlled environment chambers at constant 23 °C.** The *Miscanthus* genotypes included 28 *M. sinensis*, 2 *M. floridulus*, 2 diploid *M. ×giganteus*, and 1 triploid *M. ×giganteus*. The genotypes were evaluated for response to three day length treatments: 15 h (orange data), 12.5 h (green data) and 10 h (blue data), respectively. Collection sites of the genotypes obtained from the wild are shown by their placement on the geographic map. *Miscanthus* genotype names are printed in colors representing six *M. sinensis* genetic groups identified by Clark *et al.* [23, 24], which included Korea/North China (red), Yangtze-Qinling (green), Northern Japan (blue), Southern Japan (yellow), Sichuan Basin (orange), and Southeastern China plus tropical (purple); for interspecific hybrids between *M. sacchariflorus* and *M. sinensis*, the dominant *M. sinensis* genetic group is shown. The inset boxplots depict variation among and within the three day-length treatments; treatments labeled with the same letter were not significantly different based on Tukey's HSD test at α = 0.05. The inset regression plots show linear relationships between traits and absolute values of latitude at collection sites for the 28 *Miscanthus* genotypes with geographical information. Note that under 15 h days culm length was greatest and only weakly associated with latitude of origin, whereas culm length shortest under 10 h days but strongly associated with latitude of origin. Also note that accessions from central and northern Japan had fewer leaves under 10 and 12.5 h than at 15 h; in contrast, accessions from similar latitudes in China when grown under short days had similar or greater numbers of leaves as under long days, yet the accessions

from China and Japan both had short culms when grown under short days, indicating different mechanisms of responding to day length resulting in similar height outcomes.

photoperiod of 1 min d$^{-1}$ from 12.5 h to 11 h over the course of 3 months, then holding a constant 11 h day length for an additional ~2 months. Sugarcane is difficult to flower and synchronize for crosses, so sugarcane breeders commonly use photoperiod facilities to induce flowering by an initial exposure to ~12.5 days followed by a declining day length of 30–60 sec d$^{-1}$ [18, 19, 29, 30]. Further improvements in the number of genotypes that can be flowered in our greenhouse might be obtained by adjusting the rate of decline in photoperiod. Recently, two studies found that a photoperiod decline of 40–45 sec d$^{-1}$ was likely superior to 30 or 60 d$^{-1}$ for flowering most sugarcane genotypes [31, 32].

The early establishment of the sugarcane pots in Expt. 1a relative to Expts. 1b and 1c was advantageous, resulting in more than twice as many genotypes flowering in autumn and early winter, and also enabling a second flush of flowering for some genotypes in late winter and spring that was not obtained in the later-planted experiments. Julian *et al*. [33] and Berding [19] observed that the optimal age of sugarcane stems for floral induction was 12–16 weeks. In our study, when the critical 12.5 h photoperiod was reached in mid-September, the age of the

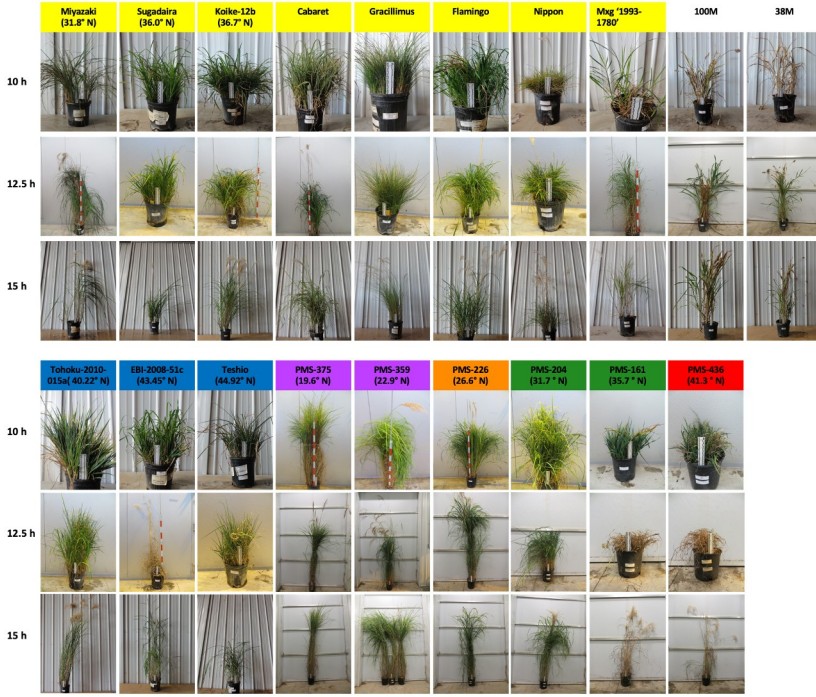

**Fig 6. Photographs of plants at the end of the growth chamber experiments on the effect of day-length on *Miscanthus*.** Plants were tested under each of three day lengths: 10, 12.5, and 15 h. Colored background behind *Miscanthus* genotype names represent the *M. sinensis* genetic groups identified by Clark *et al*. [23, 24], which included Korea/North China (red), Yangtze-Qinling (green), Northern Japan (blue), Southern Japan (yellow), Sichuan Basin (orange), and Southeastern China plus tropical (purple); for interspecific hybrids between *M. sacchariflorus* and *M. sinensis* (Nippon and M×g '1993–1780'), the dominant *M. sinensis* genetic group is shown. Representatives of each genetic group and a range of latitudes (in parentheses) are shown. In each photo, plant size is scaled by either a 20 cm ruler (black and white) or a 1 m stick (orange and white). Note that accessions originating from high latitudes typically remained short and had few or no flowering stems when grown under short days but were taller and flowered when grown under long days.

sugarcanes was ~20 weeks for Expt. 1a, ~14 weeks for Expt. 1b, and 6 weeks for Expt. 1c. Thus, under our conditions, an establishment phase about six weeks longer than the ~14 weeks optimum previously reported was beneficial. Though the later planting of sugarcane in Expts. 1b and 1c helped limit height, thereby avoiding stems reaching the roof of a greenhouse with 6.1 m side-walls, the height problem could be better addressed by air layering stems so that they could be cut if they get too tall, without sacrificing growth. Air layering would also make it easier for workers to move stems during flowering to facilitate emasculation and crossing.

Species and genotype also had a large effect on timing and ease of flowering of sugarcane in our study. The earliest flowering species were *S. spontaneum* and *S. arundinaceum*, which was expected [34]. *Saccharum* hybrids with a high proportion ancestry from *S. spontaneum*, such as 'L79-1002', 'Ho06-9001', and 'Ho06-9002', were among the most consistent to flower in our study. However, some commercial sugarcane materials, such as 'L09-105', also flowered well in our study.

## Effects of day length on *Miscanthus* development

Photoperiod profoundly affected all aspects of *Miscanthus* growth and development that we studied, especially flowering. Expt. 2 demonstrated that few *M. sinensis* or *M. floridulus* genotypes that originated outside of the tropics flowered well under 12.5 h days or less, yet all the subtropical and temperate-sourced genotypes flowered well under 15 h days (Fig 3), which is the photoperiod during the summer solstice at 40˚ N, where Urbana is located. Jensen *et al.* [13] concluded that *M. sacchariflorus* is a quantitative short-day plant because flowering under a constant 12.5 h or a declining photoperiod mimicking 34.1˚ N was >50 days earlier than for those grown under constant 15.3 h days, which was generally consistent with our observations for *M. sinensis* in Expt. 2, though critical photoperiods may vary by species and genotype. For *M. sacchariflorus* grown under a declining photoperiod mimicking 34.1˚ N, Jensen *et al.* [13] estimated that floral induction occurred between 13.8 and 12.5 h day lengths.

Notably, Jensen *et al.* [13] also observed that *M. sacchariflorus* genotypes originating from 34.5˚ N and higher failed to flower under a declining photoperiod mimicking 34.1˚ N, even though some produced flag leaves when day lengths were between 12.7 and 12.1 h; in contrast, *M. sacchariflorus* genotypes from lower latitudes flowered when days were shorter than 12 h. For *M. sinensis*, we similarly observed that flowering of genotypes from temperate latitudes (>34˚ N) was inhibited by short days (constant 10 and 12.5 h), even though some produced flag leaves, whereas flowering was consistently strong under 15 h days. In addition to not flowering, *M. sinensis* from temperate latitudes produced many short culms under 10 and 12.5 h days, resulting in a short and dense morphology similar to that of many alpine plants (Figs 5 and 6, S1 and S2 Figs). Such a dense and short morphology can protect apical meristems from freeze damage by keeping them below the soil surface, and limit water loss by reducing air flow around leaves. Thus, for *Miscanthus*, relatively short days can accelerate floral induction, but below a critical threshold, especially for genotypes adapted to high latitudes, short days can signal that plants should prepare for winter, and importantly this response is epistatic to flowering. Similarly, short-days (<12.5 h) have been shown to induce dormancy and reduce or prevent flowering in switchgrass (*Panicum virgatum*) and big bluestem (*Andropogon gerardii*) (especially for high-latitude populations), which are also quantitative short-day, perennial, $C_4$ grasses [35–37]. Moreover, low-intensity light extension of day length prevented or reversed this dormancy in switchgrass [38].

In the greenhouse experiment (Expt. 1), we established *Miscanthus* plants at different times (implemented by different initial planting dates, by cutting back established plants, or by cutting back plants then storing them at 4 ˚C for 1 month to mimic dormancy) in an effort to identify treatments that could delay flowering sufficiently to synchronize with sugarcane, but time of

establishment was only effective if day length was conducive. Establishing *Miscanthus* plants from March through the first week of July enabled genotypes from subtropical and temperate latitudes to flower in late summer and early autumn (Fig 1; Expts. 1a and 1b), indicating that floral induction occurred during photoperiods greater than 12.5 h, prior to mid-September, which was consistent with the results of Expt. 2 and Jensen *et al.* [13]. Moreover, there was little difference in flowering time between plants of the same genotype established in June compared to those established in early July (Fig 1; Expt. 1b), indicating that more rapid flowering associated with the shorter photoperiods encountered by mature stems of the later planting compensated for the difference in planting date. Thus, when established in spring and early summer, the *Miscanthus* genotypes from subtropical and temperate latitudes flowered early and failed to synchronize with most of the sugarcane genotypes, though some overlap was achieved with the early-flowering *S. spontaneum* and *S. arundinaceum* accessions. With early-season establishment and under the declining photoperiod treatment during autumn in the greenhouse, only the two tropical *Miscanthus* genotypes tested (*M. floridulus* 'US56-002-03' and *M. sinensis* 'PMS-375') flowered late enough to consistently synchronize flowering with the first flush of sugarcane flowering in Expt. 1a (in late November and early December) and the single flush of sugarcane flowering in Expts. 1b and 1c (Fig 1), which was consistent with the results of Expt. 2 that these low-latitude genotypes flowered strongly under constant 12.5 h days but did not flower under 15 days (Fig 3). When *Miscanthus* genotypes from subtropical and temperate latitudes were established during the last week of July or later in the summer or autumn, few flowered because the photoperiod was too short to be conducive by the time stems had sufficiently matured; the exceptions were primarily *M. sacchariflorus* genotypes, and the tropical *M. floridulus* 'US56-002-03' and *M. sinensis* 'PMS-375' (Fig 1; Expts. 1a-c). For example, when some *M. sacchariflorus* genotypes were established during the first week of September, flowering was delayed until November, which would allow synchronization with many sugarcane genotypes (Fig 1; Expt. 1c).

## Synchronizing flowering time of sugarcane and *Miscanthus* to facilitate intergeneric crosses

To synchronize flowering of sugarcane and *Miscanthus* in the autumn, it would be advantageous to hasten flowering of the sugarcane and delay flowering of the *Miscanthus*. Furthermore, it would be desirable to promote flowering of both genera during the late winter and spring. To achieve strong flowering of sugarcane, in a high-latitude greenhouse such as ours, during autumn and early winter, and promote flowering in spring, the plants should be established from cuttings five to six months prior to onset of the 12.5 h and declining day lengths critical for floral induction.

For *Miscanthus* that originated from the tropics, the same environment that is conducive to flowering of sugarcane, including declining photoperiod, will likely result in synchronized flowering between the two genera during the late autumn. Moreover, cutting back established plants of tropical *Miscanthus* genotypes in early September, December or January can be used to delay flowering and synchronize with a second spring flush of sugarcane flowering. We note, however, that cold treatments after cutting were disadvantageous for flowering tropical *Miscanthus* genotypes.

For *M. sinensis* genotypes that originated from subtropical and temperate latitudes, however, the short and declining day lengths needed to flower sugarcane are not conducive to synchronization of flowering between the two genera. One strategy for synchronizing the flowering of subtropical and temperate *M. sinensis* is to grow the plants under a conducive photoperiod, such as constant 15 h days (in controlled environment chambers or in a different greenhouse than that used to grow the sugarcanes) and use empirical data on the number of

growing days needed to obtain first or peak flowering (e.g. S1–S3 Tables) to choose a planting date that would achieve concurrent flowering with sugarcane in late autumn and early winter or in spring. Though data from Expt. 2 indicated that a constant 15 h day length should facilitate strong flowering after a defined number of days for most if not all subtropical and temperate *M. sinensis*, it may not be the fastest or optimal day length. Given that 12.5 h days was observed to be too short, an optimal day length for flowering subtropical and temperate *M. sinensis* may be between 13 and 15 h, though further testing would be needed to determine this. Moreover, Castro *et al.* [37] found that providing switchgrass, a cumulative short-day plant, with 24 h photoperiod, resulted in multiple rounds of flowering and this could be used to synchronize flowering between early and late genotypes. Given these promising results from switchgrass and the high level of flowering observed under ~15 h days in *M. sinensis* (Expt. 2) and *M. sacchariflorus* [13], it would be worthwhile to investigate if a 24 h photoperiod would also produce sequential flowering in *Miscanthus*.

For *M. sacchariflorus* grown under the short and declining photoperiod needed to flower sugarcane, most genotypes flowered as late as the end of October, which was still too early to synchronize with most sugarcane genotypes. However, *M. sacchariflorus* ssp. *lutarioriparius* 'PF30022' was a notable exception, in that plants given a cut plus 1 month cold treatment in September, December or January then grown under the short and declining day length regime that was conducive to flowering sugarcane, produced flowers in late November or March/April, which would match well with sugarcane flowering (Fig 1; Expt. 1a). *M. sacchariflorus* ssp. *lutarioriparius* is indigenous to the lower Yangtze River watershed and is a tall plant with high-biomass yield that is harvested locally to produce paper on a commercial scale [3, 39, 40], so crossing it to sugarcane would be desirable. However, to delay flowering of most *M. sacchariflorus* genotypes for synchronization with sugarcane, we suggest cultivation of the former under a constant conducive photoperiod for an empirically determined amount of time, similar to the strategy we propose for subtropical and temperate *M. sinensis*. However, there is currently little information on what might be optimal photoperiods for flowering *M. sacchariflorus*. Jensen *et al.* [13] observed that *M. sacchariflorus* flowered under constant 15.3 h days, so that would be one option. We observed that under constant 13 h days, three out of six *M. sacchariflorus* genotypes from eastern Russia planted during the first week of October began to flower by early December (Fig 1, Expt. 1c), which would be suitably late for crossing with sugarcane; however, because these accessions originated from ~49˚ N, an optimal day length for flowering them might be expected to be greater than 13 h. Given that *M. sacchariflorus* originates from a wide range of latitudes, day lengths that are optimal for flowering might be expected to range from 12.5 to 16 h.

In this study, we identified barriers to synchronizing the flowering of sugarcane and *Miscanthus*, and proposed methods to circumvent these. For a given genotype of *Miscanthus*, a range of flowering dates may be obtained by staggered plantings grown under a single conducive constant day length, or by planting on a single date and growing under a range of conducive and constant day lengths, leveraging the short-day response of faster flowering under shorter day lengths than longer ones. By controlling flowering time of sugarcane and *Miscanthus*, plant breeders will be better able to improve these crops via intra- and intergeneric crosses of their choosing.

## Supporting information

**S1 Fig. Photographs of *Miscanthus* from the Southeastern China plus tropical group at the end of the growth chamber experiments on the effect of day-length on *Miscanthus*.** Plants were tested under each of three day lengths: 10, 12.5, and 15 h. Colored background behind

*Miscanthus* genotype names represent the *M. sinensis* genetic groups identified by Clark *et al.* [23, 24], which included Korea/North China (red), Yangtze-Qinling (green), Northern Japan (blue), Southern Japan (yellow), Sichuan Basin (orange), and Southeastern China plus tropical (purple). In each photo, plant size is scaled by either a 20 cm ruler (black and white) or a 1 m stick (orange and white).
(TIF)

**S2 Fig. Photographs of *Miscanthus* from China and Japan at the end of the growth chamber experiments on the effect of day-length on *Miscanthus*.** Plants were tested under each of three day lengths: 10, 12.5, and 15 h. Colored background behind *Miscanthus* genotype names represent the *M. sinensis* genetic groups identified by Clark *et al.* [23, 24], which included Korea/North China (red), Yangtze-Qinling (green), Northern Japan (blue), Southern Japan (yellow), Sichuan Basin (orange), and Southeastern China plus tropical (purple); for interspecific hybrids (PMS-300) between *M. sacchariflorus* and *M. sinensis*, the dominant *M. sinensis* genetic group is shown. In each photo, plant size is scaled by either a 20 cm ruler (black and white) or a 1 m stick (orange and white). Note that accessions originating from high latitudes typically remained short and had few or no flowering stems when grown under short days but were taller and flowered when grown under long days.
(TIF)

**S1 Table. First flowering date of *Miscanthus* and sugarcane in 2014–2015 greenhouse experiment.**
(XLSX)

**S2 Table. First flowering date of *Miscanthus* and sugarcane in 2015–2016 greenhouse experiment.**
(XLSX)

**S3 Table. First flowering date of *Miscanthus* and sugarcane in 2016–2017 greenhouse experiment.**
(XLSX)

**S4 Table. Trait summary statistics in the controlled growth chamber experiment.**
(XLSX)

**S5 Table. Raw data of the greenhouse experiment (2014–2017).**
(XLSX)

**S6 Table. Raw data of the controlled growth chamber experiment.**
(XLSX)

## Acknowledgments

We thank Benjamin Baechle, Colten Maertens, Helen Gapsis, Homayoun Watan, Melina Salgado, Walker Maffit, and Gyu Hoi Cha for technical assistance.

## Author Contributions

**Conceptualization:** Erik J. Sacks.

**Data curation:** Hongxu Dong, Erik J. Sacks.

**Formal analysis:** Hongxu Dong.

**Funding acquisition:** Stephen P. Long, Erik J. Sacks.

**Investigation:** Hongxu Dong, Erik J. Sacks.

**Methodology:** Hongxu Dong, Erik J. Sacks.

**Project administration:** Stephen P. Long, Erik J. Sacks.

**Resources:** Xiaoli Jin, Kossonou Anzoua, Larisa Bagmet, Pavel Chebukin, Elena Dzyubenko, Nicolay Dzyubenko, Bimal Kumar Ghimire, Kweon Heo, Douglas A. Johnson, Hironori Nagano, Andrey Sabitov, Junhua Peng, Toshihiko Yamada, Ji Hye Yoo, Chang Yeon Yu, Hua Zhao, Stephen P. Long, Erik J. Sacks.

**Software:** Hongxu Dong, Lindsay V. Clark.

**Supervision:** Erik J. Sacks.

**Validation:** Hongxu Dong.

**Visualization:** Hongxu Dong, Lindsay V. Clark.

**Writing – original draft:** Hongxu Dong.

**Writing – review & editing:** Hongxu Dong, Erik J. Sacks.

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
