## [Decision Letter · Decision Letter 0]

21 Oct 2020

PONE-D-20-30190

Manage flowering time in Miscanthus and sugarcane to facilitate intra- and intergeneric crosses

PLOS ONE

Dear Dr. Sacks,

Thank you for submitting your manuscript to PLOS ONE. After careful consideration, we feel that it has merit but does not fully meet PLOS ONE’s publication criteria as it currently stands. Therefore, we invite you to submit a revised version of the manuscript that addresses the points raised during the review process.

We look forward to receiving your revised manuscript.

Kind regards,

Tzen-Yuh Chiang

Academic Editor

PLOS ONE

Journal Requirements:

2)  We note that Figures 3, 4 and 5 in your submission contain map/satellite images which may be copyrighted. All PLOS content is published under the Creative Commons Attribution License (CC BY 4.0), which means that the manuscript, images, and Supporting Information files will be freely available online, and any third party is permitted to access, download, copy, distribute, and use these materials in any way, even commercially, with proper attribution. For these reasons, we cannot publish previously copyrighted maps or satellite images created using proprietary data, such as Google software (Google Maps, Street View, and Earth). For more information, see our copyright guidelines: http://journals.plos.org/plosone/s/licenses-and-copyright.

a) .    You may seek permission from the original copyright holder of Figure(s) [#] to publish the content specifically under the CC BY 4.0 license.

b).    If you are unable to obtain permission from the original copyright holder to publish these figures under the CC BY 4.0 license or if the copyright holder’s requirements are incompatible with the CC BY 4.0 license, please either i) remove the figure or ii) supply a replacement figure that complies with the CC BY 4.0 license. Please check copyright information on all replacement figures and update the figure caption with source information. If applicable, please specify in the figure caption text when a figure is similar but not identical to the original image and is therefore for illustrative purposes only.

Reviewers' comments:

Reviewer's Responses to Questions

**Comments to the Author**

1. Is the manuscript technically sound, and do the data support the conclusions?

Reviewer #1: Yes

Reviewer #2: Yes

2. Has the statistical analysis been performed appropriately and rigorously? 

Reviewer #1: Yes

Reviewer #2: Yes

3. Have the authors made all data underlying the findings in their manuscript fully available?

Reviewer #1: Yes

Reviewer #2: Yes

4. Is the manuscript presented in an intelligible fashion and written in standard English?

Reviewer #1: Yes

Reviewer #2: Yes

5. Review Comments to the Author

Reviewer #1: In this manuscript entitled " Manage flowering time in Miscanthus and sugarcane to facilitate intra- and intergeneric crosses” to synchronize flowering time of Saccharum and Miscanthus species by using different cultural treatments. They found the barriers to synchronizing the flowering of sugarcane and Miscanthus, and also proposed methods. For Miscanthus, flowering dates may be obtained by staggered plantings grown under a single conducive constant day length, or by planting on a single date and growing under conducive and constant day lengths. The important application of the article is used to improve breeding program of crops by controlling flowering time of sugarcane and Miscanthus. In general, this paper is technically accurate, and interesting. I suggest the authors do some minor revisions, before accepting.

Minor comments

1. Lin 276-277 The first flowering flush was observed from October 2014 to January 2015, with… Please recheck, October 2014 to Dec 2014?

2. L307 from “5” August 2015 to “19” December 2015 in supplementary table.

3. L434 to 30 or 60 sec d-1

4. It is redundant in discussion paragraph, I suggest that authors reduce the length.

Reviewer #2: The topic is interesting. Terminology and methods they used in this study are clearly described and acceptable in a scientific journal. The authors present an interesting publication that reports the usefulness of hybridization based on manage flowering time in Miscanthus and sugarcane. The aim of this manuscript is interesting and important for the hybridization based on manage flowering time in Miscanthus and sugarcane. I don’t have any negative recommendations on this manuscript and I will suggest accepting this manuscript in the current form.

6. PLOS authors have the option to publish the peer review history of their article (what does this mean?). If published, this will include your full peer review and any attached files.

Reviewer #1: No

Reviewer #2: No

---

## [Author Response · Author response to Decision Letter 0]

17 Nov 2020

We greatly appreciate the comments from two reviewers and the editor. We have made revisions based on reviewer’s comments. We replaced the original Figs 3, 4, and 5 with new figures. The geographic background of original Figs 3-5 was made using ArcGIS. For the new Figs 3-5, we made the geographic background using the R package ggmap (Kahle & Wickham, 2013), which is a free, non-copyrighted, open-source programming package. I wrote my own R codes to develop these three new figures. Therefore, no copyright related issues should exist. I have also made my R codes used in this study publicly available on GitHub (https://github.com/hxdong-genetics/Geographic-map-in-Miscanthus-flowering-study).

Reference

Kahle D, Wickman H. ggmap: Spatial visualization with ggplot2. 2013. The R Journal. 5: 144-161

Reviewer #1: In this manuscript entitled " Manage flowering time in Miscanthus and sugarcane to facilitate intra- and intergeneric crosses” to synchronize flowering time of Saccharum and Miscanthus species by using different cultural treatments. They found the barriers to synchronizing the flowering of sugarcane and Miscanthus, and also proposed methods. For Miscanthus, flowering dates may be obtained by staggered plantings grown under a single conducive constant day length, or by planting on a single date and growing under conducive and constant day lengths. The important application of the article is used to improve breeding program of crops by controlling flowering time of sugarcane and Miscanthus. In general, this paper is technically accurate, and interesting. I suggest the authors do some minor revisions, before accepting.

We appreciate the reviewer’s meticulous review on our manuscript. We have made minor revisions based on reviewer’s comments. 

Minor comments

1. Lin 276-277 The first flowering flush was observed from October 2014 to January 2015, with… Please recheck, October 2014 to Dec 2014?

We incorporated the reviewer’s comment.

2. L307 from “5” August 2015 to “19” December 2015 in supplementary table.

We incorporated the reviewer’s comment.

3. L434 to 30 or 60 sec d-1

We double checked the references, previous studies used declining day length ranging from 30 sec to 60 sec per day, not just 30 sec or 60 sec per day. Therefore, we need to keep our original writing.

4. It is redundant in discussion paragraph, I suggest that authors reduce the length.

We think that it is important to keep the current discussion intact. Our study consists of multiple experiments with multiple years. It would be vital to appropriately repeat key information in discussion so that readers could keep track the findings of each experiment.

Reviewer #2: The topic is interesting. Terminology and methods they used in this study are clearly described and acceptable in a scientific journal. The authors present an interesting publication that reports the usefulness of hybridization based on manage flowering time in Miscanthus and sugarcane. The aim of this manuscript is interesting and important for the hybridization based on manage flowering time in Miscanthus and sugarcane. I don’t have any negative recommendations on this manuscript and I will suggest accepting this manuscript in the current form.

We appreciate the reviewer’s kind comments.

---

## [Decision Letter · Decision Letter 1]

30 Nov 2020

Managing flowering time in Miscanthus and sugarcane to facilitate intra- and intergeneric crosses

PONE-D-20-30190R1

Dear Dr. Sacks,

We’re pleased to inform you that your manuscript has been judged scientifically suitable for publication and will be formally accepted for publication once it meets all outstanding technical requirements.

Kind regards,

Tzen-Yuh Chiang

Academic Editor

PLOS ONE

Additional Editor Comments (optional):

Reviewers' comments:

Reviewer's Responses to Questions

**Comments to the Author**

1. If the authors have adequately addressed your comments raised in a previous round of review and you feel that this manuscript is now acceptable for publication, you may indicate that here to bypass the “Comments to the Author” section, enter your conflict of interest statement in the “Confidential to Editor” section, and submit your "Accept" recommendation.

Reviewer #1: (No Response)

Reviewer #2: All comments have been addressed

2. Is the manuscript technically sound, and do the data support the conclusions?

Reviewer #1: (No Response)

Reviewer #2: Yes

3. Has the statistical analysis been performed appropriately and rigorously? 

Reviewer #1: (No Response)

Reviewer #2: Yes

4. Have the authors made all data underlying the findings in their manuscript fully available?

Reviewer #1: (No Response)

Reviewer #2: Yes

5. Is the manuscript presented in an intelligible fashion and written in standard English?

Reviewer #1: (No Response)

Reviewer #2: Yes

6. Review Comments to the Author

Reviewer #1: (No Response)

Reviewer #2: All comments have been addressed and I don't have more comment. I will suggest accepting this manuscript in the current form.

7. PLOS authors have the option to publish the peer review history of their article (what does this mean?). If published, this will include your full peer review and any attached files.

Reviewer #1: No

Reviewer #2: No

---

## [Editor Report · Acceptance letter]

23 Dec 2020

PONE-D-20-30190R1 

Managing flowering time in *Miscanthus* and sugarcane to facilitate intra- and intergeneric crosses 

Dear Dr. Sacks:

I'm pleased to inform you that your manuscript has been deemed suitable for publication in PLOS ONE. Congratulations! Your manuscript is now with our production department. 

Kind regards, 

on behalf of

Dr. Tzen-Yuh Chiang 

Academic Editor

PLOS ONE